# Bardet-Biedl syndrome proteins modulate the release of bioactive extracellular vesicles

Ann-Kathrin Volz [1], Alina Frei [1], Viola Kretschmer[1], António M. de Jesus Domingues [2], Rene F. Ketting [2,3], Marius Ueffing[4], Karsten Boldt [4], Eva-Maria Krämer-Albers [3] & Helen L. May-Simera [1✉]

Primary cilia are microtubule based sensory organelles important for receiving and processing cellular signals. Recent studies have shown that cilia also release extracellular vesicles (EVs). Because EVs have been shown to exert various physiological functions, these findings have the potential to alter our understanding of how primary cilia regulate specific signalling pathways. So far the focus has been on lgEVs budding directly from the ciliary membrane. An association between cilia and MVB-derived smEVs has not yet been described. We show that ciliary mutant mammalian cells demonstrate increased secretion of small EVs (smEVs) and a change in EV composition. Characterisation of smEV cargo identified signalling molecules that are differentially loaded upon ciliary dysfunction. Furthermore, we show that these smEVs are biologically active and modulate the WNT response in recipient cells. These results provide us with insights into smEV-dependent ciliary signalling mechanisms which might underly ciliopathy disease pathogenesis.

[1] Institute of Molecular Physiology, Johannes Gutenberg-University, Mainz, Germany. [2] Institute of Molecular Biology, Mainz, Germany. [3] Institute of Developmental Biology and Neurobiology, Johannes Gutenberg-University, Mainz, Germany. [4] Medical Bioanalytics, Institute for Ophthalmic Research, Eberhard-Karls University, Tübingen, Germany. ✉email: may-simera@uni-mainz.de

The primary cilium is a highly specialised microtubule-based signalling organelle that extends from the cell membrane of most eukaryotic cell types. In the past decade, an explosion of data has linked the primary cilium to several crucial cellular processes and various signalling pathways (reviewed in ref. [1]). The ciliary membrane, although continuous with the plasma membrane, is a specialised compartment that selectively and dynamically accommodates a multitude of receptors and channels. As a central organelle for signal transduction, the primary cilium is linked to several signalling pathways, including Hedgehog (Hh), transforming growth factor β (TGF-β) and wingless (WNT) signalling[2,3]. Disruptions in ciliary structure and downstream signalling cascades lead to numerous syndromic and non-syndromic disorders with a broad range of phenotypes, collectively termed ciliopathies[4]. Bardet–Biedl syndrome is considered an archetypical ciliopathy since patients exhibit all ciliopathy phenotypes[5]. BBS proteins are important for ciliary trafficking, mutants of which are therefore ideal model systems to elucidate general ciliary functions. Eight BBS proteins (BBS1/2/4/5/7/8/9/18) form the BBSome, a protein complex that acts as an adaptor to intraflagellar trafficking and regulates ciliary signalling cascades[6]. A further three (BBS6/10/12) form the chaperonin complex, required for the correct formation of the BBSome[7].

The primary cilium was initially understood as a signalling antenna capable of receiving incoming signals and transducing this information into a cellular response. However, recently it has been shown that primary cilia are also able to transmit signals via the release of extracellular vesicles (EVs)[8–12]. EV-mediated cellular communication is emerging as a significant mechanism of cell signalling, particularly when communication between non-adjacent cells over large distances is required[13,14].

To overcome the challenges of instability and insolubility in the extracellular milieu, signalling molecules are packaged into lipid bi-layered extracellular vesicles. This enables long-range trafficking and activation of downstream signalling cascades in target cells or tissues. EVs contain a specific subset of biologically active materials, including lipids, proteins, RNA and in some instances also DNA[13]. After the transfer of EV cargo, various post-transcriptional and signalling events are initiated in recipient cells, thereby modulating developmental homoeostatic or pathological processes. EVs can be categorised based on size and mode of biogenesis[15]. Small EVs (smEVs; size <200 nm) comprise small microvesicles shedding from the plasma membrane and exosomes of endocytic origin, released into the extracellular space on the exocytosis of multivesicular bodies (MVBs)[16,17]. In contrast, larger EVs (lgEVs; >200 nm also referred to as ectosomes or microvesicles) are assembled and released directly at the plasma membrane.

Primary cilia are evolutionarily conserved sites of EV production although studies in different species have shown variations in ciliary EV biogenesis and shedding[8,11,18–20]. Their possible functions include signalling[19,21], waste disposal[22] and cilia morphogenesis[20,23,24]. Recent in vivo work in *C. elegans* has shown that ciliary EVs are released in a mechanoresponsive manner and can be directly transferred from one organism to another[25]. Due to spatial restraints and gating at the ciliary transition zone, MVBs are unlikely to enter the cilium. Therefore, ciliary derived EVs are likely to be lgEVs (ectosomes), formed by budding of the ciliary membrane. This can happen at the base of the cilium, along the lateral length of the cilium or from the apical tip and is likely to be species or cell type-dependent (reviewed in ref. [12]). In mammalian cells, it is thought that EV shedding is most likely to occur at the apical tip[11,19,22,24,26–28]. Various imaging techniques have identified vesicles directly attached to the ciliary membrane but it is not clear whether these are being released or absorbed at these sites[29–31]. Although ciliary EVs have

been gaining attention in recent years, the focus has been on lgEVs budding directly from the ciliary membrane. An association between cilia and MVB-derived smEVs has not yet been described.

Recent literature highlights a prominent role of smEVs in intercellular communication[32,33]. They have been found to traffic a subset of WNT molecules and thereby modulate WNT activity in donor and/or recipient cells in some cases over large distances (reviewed in ref. [34]). This ciliary associated signalling pathway is crucial for the development and homoeostasis of tissues and organs (reviewed in ref. [1]). Numerous cilia mutant cell and animal models exhibit defective WNT signalling[35–42]; however, the precise mechanisms underlying this are not clear. Numerous studies both in vitro and in vivo, in a variety of different animal models, have shown that increased canonical WNT signalling underlies renal cystogenesis[43–45]. Renal dysfunction is the most common and life-threatening ciliopathy phenotype (reviewed in refs. [46,47]) and the most important polycystic kidney disease proteins (PKD1 and PKD2) which predominantly localise to the ciliary membrane[48,49] have also been found in urinary derived smEVs[29].

In this study, we set out to explore smEV release in ciliated mammalian cells to test whether this can modify signalling pathways in both secreting and target cells. Because kidney disease is one of the most common ciliopathy phenotypes in patients and the largest contributor to mortality[47], we used a kidney-derived cell line. We were able to separate EV populations from immortalised kidney medullary (KM) cell lines isolated from *Bbs4* and *Bbs6* knockout mice. Bbs4 is a component of the BBSome complex required for ciliary trafficking while Bbs6 (Mkks) is a component of the BBS chaperonin complex required for BBSome assembly. Loss of either gene results in ciliary trafficking defects and *Bbs4* and *Bbs6* KM cell lines have been extensively characterised and recapitulate the cilia phenotype of related cilia knockout or knockdown cell lines[42,50]. Since the loss of BBS protein function results in defective ciliary trafficking and disruption of downstream signalling events, the *Bbs4* and *Bbs6* KM cell lines offer a unique opportunity to study molecular mechanisms associated with ciliary extracellular vesicles in vitro.

Our findings offer insights into the physiological mechanisms underlying EV generation and function. We identified an abundance of Wnt-related molecules selectively secreted via distinct modes of EV biogenesis coordinated via the cilium. This suggests that cilia do not only act as a sensory antenna but also as possible regulators of EV release. Furthermore, we found that ciliary related smEVs are taken up by target cells and initiate a biological response. The observation that ciliary dysfunction changes the mode of EV release could alter our perception of ciliary signalling mechanisms. This in turn has an impact on our understanding of downstream disease pathogenicity in all affected organs and tissues.

## Results

**Ciliary mutant cells differentially release smEVs.** We isolated extracellular vesicles via differential centrifugation from the supernatant of ciliated control and *Bbs* mutant KM cells grown under serum-free conditions. Omission of serum not only induced ciliation by halting cell cycle progression but also minimised small particle contamination. smEV pellets were characterised using transmission electron microscopy (TEM), western blot and nanoparticle tracking analysis (NTA). TEM micrographs from all three cell lines revealed spherical biconcave structures 100–150 nm in size, typical of smEVs (Fig. 1a). Western blot analysis using antibodies against common EV markers revealed signals for Tsg101, Flot-1, CD81 and CD9 in the secreted smEV pellets (Fig. 1b), which implied that our

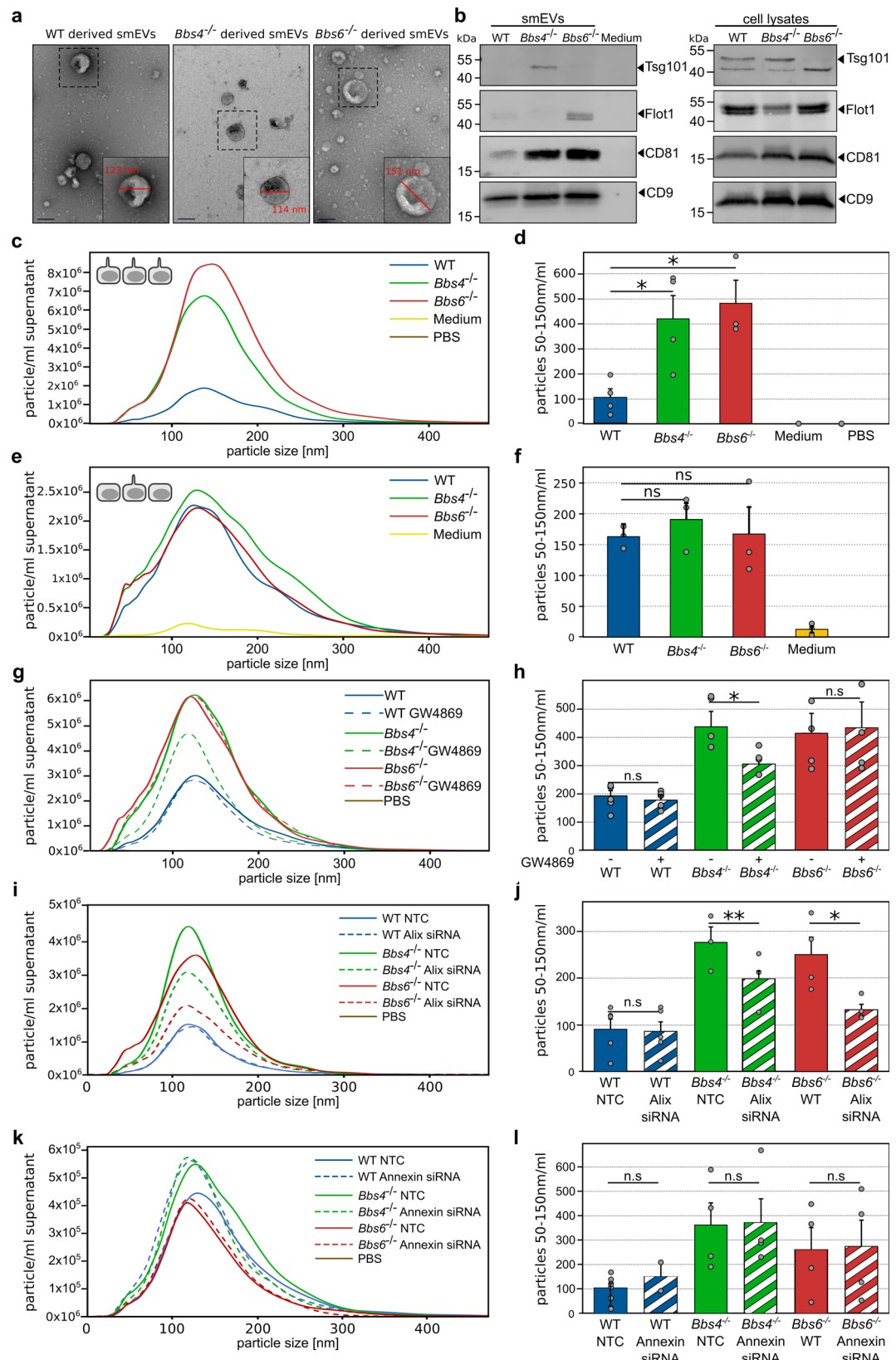

method of preparation enriched for EVs. We noticed distinct differences in protein composition between the smEV pellet from control and ciliary mutant cells. Tsg101, a protein important during sorting endocytic cargo into MVBs[51,52], could only be detected in smEVs secreted from $Bbs4^{-/-}$ cells. Flotillin-1, a

membrane-raft associated protein important for endosomal trafficking, was more readily detected in $Bbs6^{-/-}$ smEVs but was absent from $Bbs4^{-/-}$ smEVs. Flotilin is involved in the recognition of ubiquitinated cargo by ESCRT-0 and -I and serves as a ubiquitin-based sorting platform in EV cargo sorting[53,54]. The

**Fig. 1 Characterisation of control and ciliary mutant extracellular vesicles.** Extracellular vesicles (EVs) were isolated from ciliated kidney medullary (KM) cell culture supernatant using differential ultracentrifugation. **a** Morphological properties of isolated EVs were determined via negative staining followed by transmission electron microscopy and revealed 100–150 nm vesicles (scale bars: 100 nm; experiment was repeated two times). **b** Western blot analysis identified known EV markers (Tsg101, Flot-1, CD81 and CD9). We observed differences in protein expression between samples. The experiment was repeated three times. Original blots are provided as the source data file. **c, d** Nano Particle Tracking Analysis (NTA) measurements from ciliated wild-type cells show less smEV release compared to ciliated $Bbs4^{-/-}$ and $Bbs6^{-/-}$ cells ($_N$WT, $Bbs4^{-/-}$, medium, PBS = four independent experiments; $_N Bbs6^{-/-}$ = three independent experiments; unpaired $t$ test pWT vs. $Bbs4^{-/-}$ = 0.038; pWT vs. $Bbs6^{-/-}$ = 0.043). Most particles were between 50 and 200 nm in size. Medium and PBS control contained no particles. **e, f** NTA measurements from serum-fed KM cells showed less vesicles. It was no difference in particle number between WT, $Bbs4^{-/-}$ and $Bbs6^{-/-}$ observed. ($_N$WT, $Bbs4^{-/-}$, $Bbs6^{-/-}$, medium = three independent experiments; unpaired $t$ test pWT vs. $Bbs4^{-/-}$ = 0.382; $pWT^{-/-}$ vs $Bbs6^{-/-}$ = 0.927; $pBbs4^{-/-}$ vs. $Bbs6^{-/-}$ = 0.669). **g, h** NTA of KM cells treated with GW4869 showed a decrease in particle number in $Bbs4^{-/-}$ samples compared to untreated $Bbs4^{-/-}$ KM cells. WT and $Bbs6^{-/-}$ vesicle number was not reduced upon treatment ($_N$WT = five independent experiments; $_N Bbs4^{-/-}$, $Bbs6^{-/-}$ = four independent experiments; paired $t$ test pWT = 0.748; $pBbs4^{-/-}$ = 0.02; $pBbs6^{-/-}$ = 0.542). **i, j** Alix knockdown via siRNA significantly decreased the number of vesicles released from ciliary mutant cells. EV release was unchanged in control cells upon knockdown ($_N$WT = five independent experiments; $_N Bbs4^{-/-}$ = three independent experiments; $_N Bbs6^{-/-}$ = four independent experiments; unpaired $t$ test pWT = 0.695; $pBbs4^{-/-}$ = 0.007; $pBbs6^{-/-}$ = 0.038). **k, l** EV release was unchanged in control and cilia mutant cells upon Annexin1 (Anxa1) knockdown ($_N$WT, $Bbs4^{-/-}$, $Bbs6^{-/-}$ = 3 independent experiments; unpaired $t$ test pWT=0.576; $pBbs4^{-/-}$ = 0.943; $pBbs6^{-/-}$ = 0.930). **d, f, h, j, l** Data are presented as mean values +/− SEM. n.s., $P > 0.05$, *$P \leq 0.05$, **$P < 0.01$, ***$P < 0.001$. Source data are provided as a Source Data file.

tetraspanins CD9 and CD81 were particularly enriched in ciliary mutant smEVs.

The concentration and size distribution of EVs were measured via NTA. While all cell lines excreted EVs ranging in size from 50 to 300 nm, we observed a dramatic difference in vesicle number between control and the ciliary mutant samples (Fig. 1c), despite normalisation to cell number. Calculating the number of vesicles in the smEV size range (50–150 nm) both ciliary mutant cell lines secreted over four times as many particles compared to control (Fig. 1d). Importantly, the PBS and medium controls showed no particle contamination. Upon serum starvation, over 50% of WT KM cells were ciliated (Supplementary Fig. 1a). Both $Bbs4^{-/-}$ and $Bbs6^{-/-}$ KM cells displayed significantly more ciliated cells with significantly longer cilia (Supplementary Fig. 1b, c). To determine whether the additional smEVs secreted from ciliary mutant cells were associated with ciliation, we repeated the experiment using cells grown with EV-depleted serum (Fig. 1e). The presence of serum reduced ciliation in mutant cells, but not in control cells (Supplementary Fig. 1d). Under these conditions, ciliary mutant cells secreted the same number of smEVs compared to the control (Fig. 1f), suggesting that the significant increase in smEV secretion in the mutant cell is related to disrupted ciliary functions. Importantly, these differences were not due to an increase in apoptosis, since mutant cells actually exhibited less cell death than control cells (Supplementary Fig. 2).

To address the origin of the additional vesicles, we treated the cells with GW4869, a compound that inhibits ceramide dependent intraluminal vesicle budding into multivesicular bodies (MVBs) and therefore endocytically derived smEVs[55]. NTA analysis of treated and untreated control and $Bbs6^{-/-}$ cells showed no difference in vesicle release (Fig. 1g, h). However, upon treatment of $Bbs4^{-/-}$ cells, we observed a significantly decreased release of smEVs. To further validate whether the additional vesicles released by ciliary mutant KM cells are MVB derived, we knocked down *Alix* (*Pdcd6ip*) using siRNA. Alix is a component of the endosomal sorting complex required for transport (ESCRT) machinery, involved in EV biogenesis and plays a critical role in MVB cargo sorting[56,57]. Loss of Alix has been shown to decrease the number of MVB-derived smEVs[58,59]. We verified the loss of Alix in KM cells after siRNA treatment via qPCR (Supplementary Fig. 3a). In control and $Bbs6^{-/-}$ KM cells, we achieved a knockdown of 60–70%, which was higher than in $Bbs4^{-/-}$ KM cells in which we only achieved a 40% knockdown. Despite an incomplete knockdown of Alix expression, a significantly reduced number of EVs was released in both ciliary mutant samples ($Bbs4^{-/-}$ and $Bbs6^{-/-}$, Fig. 1i, j) compared to treatment with

non-targeting control (NTC) siRNA. In contrast, there was no change in EV release in control cells. Manipulating biogenesis of lgEVs derived by direct budding of the plasma membrane via knockdown of Annexin A1 made no difference to the secretion of smEVs in either control or ciliary mutant cells (Fig. 1k, l). Annexin A1 has recently been shown to be a specific marker for lgEVs shed from the plasma membrane[60]. Combined, although these experiments do not completely rule out the shedding of smEVs directly from the ciliary axoneme, the data suggest that a proportion of excess vesicles secreted from ciliary mutant cells are actively secreted via an MVB-associated sorting mechanism. This is also supported by the increase in Tsg101, a protein important during sorting endocytic cargo into MVBs[51,52], which could only be detected in smEVs secreted from $Bbs4^{-/-}$ cells. In summary, both the NTA and western blot data indicated a difference in smEV release and composition upon loss of ciliary function.

**Ciliary EV protein composition dependent on the mode of release.** We further characterised ciliary EV protein cargo using liquid chromatography-mass spectrometry. In comparison to unciliated cells, ciliated cells from both control and mutant cell lines were loaded with significantly more protein types (Fig. 2a and Supplementary Data 1 and 2). This was consistent with our data related to changes in EV number, so we focused our attention on further characterising EVs secreted specifically from ciliated cells.

First EVs secreted from ciliated control KM cells were examined in more detail. We generated EV pellets that were enriched for either smEVs or lgEVs (prepared via ultracentrifugation at 100 or 10 K pellets, respectively; Supplementary Data 1 and 3). In addition, we also prepared mixed pellets that contained both (100 K excluding 10 K step). Each point on the scatter plot represents one identified protein, its location on the $x$ axis depicts its relative abundance (Fig. 2b). Based on their content, we could differentiate between two protein populations, those present in both smEVs and lgEVs (central cluster) and those enriched in lgEVs (left cluster). We identified 93 out of the top 100 smEV (exosomal) proteins as defined by ExoCarta (ExoCarta top 100 proteins list, www.ExoCarta.org; highlighted in blue) (Fig. 2b, e). The vast majority of these proteins were detected in the central cluster (smEV pellet/Mixed pellet).

The cluster on the left was enriched for proteins absent from the smEV pellet and therefore more commonly detected in lgEVs. In this cluster, we identified lgEV proteins that were significantly depleted by a factor of four compared to the mixed preparation ($<\log_2$ ratio-2). These proteins are thus more likely to only be

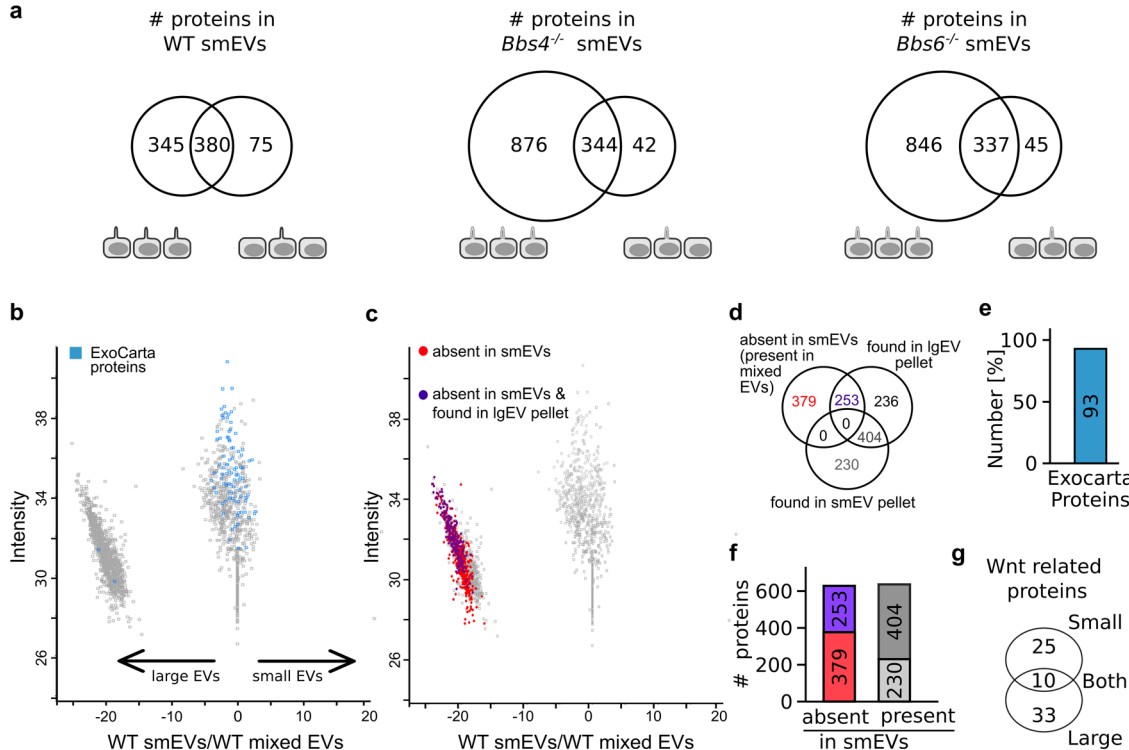

**Fig. 2 Specific proteins exhibit unique modes of EV release. a** Comparison of EV protein content identified with LC-MS from ciliated (serum-starved) and less ciliated (serum-fed) KM cells showed less proteins released from serum-fed cells. **b**, **c** Proteins identified via liquid chromatography-mass spectrometry analysis in four out of six (smEV and mixed EV) or three out of four (lgEV) pellets isolated from ciliated control KM supernatant. Each point represents one protein. Its location on the *x* axis depicts its relative abundance. **b** Proteins listed on the ExoCarta Top 100 protein list are marked in blue (exocarta.org). ExoCarta proteins were predominantly found in the middle cluster, therefore equally abundant in small vs. large EVs. **c** Proteins localising to the left cluster were considered enriched in lgEVs and absent from smEVs. Significantly enriched lgEVs (relative abundance ≤ −10 and a significance −log ≥2) are labelled red. Significantly enriched lgEVs that were also identified in separate lgEV preparations are labelled purple. **d** Absolute numbers of different protein populations depicted in (**c**, **d**). **e** In all, 93 out of the 100 ExoCarta proteins were detected. **f** Graphical representation of smEV protein abundance. **g** Absolute numbers of Wnt signalling proteins identified in different populations as determined by Uniprot or GetGo (uniport.org; getgo.russelllab.org). **b**–**f** Source data are provided as Source and Supplementary data file.

found in lgEVs (highlighted in red; Fig. 2c). Approximately 40% of these proteins (253/632) were also identified upon preparation of a pure lgEV pellet (highlighted in purple) (Fig. 2c, d). By comparing protein content from different EV preparations, we were able to categorise various populations present predominantly in smEVs, predominantly in lgEVs or in both (Fig. 2d, f).

To determine the biological relevance for these protein populations, we performed enrichment analyses using GetGo, restricting the output to cellular components and biological processes. The top GO term under cellular component associated with proteins in our smEV-only and mixed EV preparations was Extracellular Exosome with 157 (68%) and 299 (74%) of identified proteins, respectively (GO:0070062, Table 1), suggesting that our method of preparation for smEVs enriched for MVB-associated vesicles. The top GO term for cellular component associated with proteins found only in our lgEV preparation was cytosol with 305 (35%) of identified proteins (GO:0005829), suggesting an alternative method of release. Identification of biological processes associated with all of these protein populations highlighted six different signalling pathways (Table 1) with Wnt signalling featuring in both smEV and lgEV populations. Surprisingly, there was no overlap in terms of proteins related to Wnt signalling identified in these two populations (Table 3). Only one signalling pathway was uniquely enriched in smEVs namely integrin-mediated signalling (GO:0007229).

Taken together, these data suggest that proteins for certain signalling pathways become enriched in either smEVs or lgEVs.

**Cilia dysfunction alters EV protein composition**. Since we identified disparate proteins released via smEVs vs. lgEVs in control cells, we sought to examine whether loss of cilia function disrupts this segregation. When comparing protein content of smEVs from *Bbs* mutant cells vs. ciliated control cells, we noticed an enrichment of proteins present in ciliary mutant smEVs that were absent from control smEVs (Fig. 3a, b (right cluster), c (right cluster)). Intriguingly, a large proportion of these proteins (38% in Bbs4; 31% in Bbs6) were originally identified in control lgEVs only (highlighted in red and purple), but were significantly enriched in the *Bbs4* and *Bbs6* mutant smEV preparation (*t* test <0.05, ratio 0.25). This suggests that, in these mutant cells, proteins are released via smEVs that are originally destined for release via lgEVs in control cells (Fig. 3d).

Comparison of protein content between smEV preparations revealed that ciliary mutant EVs contained more than twice as many proteins compared to control (Fig. 3e and Supplementary Data 1). Of the 633 proteins identified in control smEVs, 96.5% were also found in mutant smEVs. In all, 530 proteins were present in both mutant EV subpopulations but not in control, 55 proteins were exclusively detected in *Bbs4−/−* smEVs and 86 exclusively in *Bbs6−/−*, again suggesting altered protein sorting to EVs upon ciliary dysfunction. Despite Bbs4 and Bbs6 having non-overlapping ciliary functions, this similar phenotype points to overlapping ciliary mechanisms.

To further characterise these smEV proteins, we only focused on those highly enriched (<log₂ ratio-2) in ciliary mutant

**Table 1 Gene ontology enrichment analysis (getgo.russelllab.org) for cellular components (CC) and biological processes (BP) for proteins identified in smEVs, lgEVs or both.**

| EV sample | Term | Name | Fisher-C-PV | # of proteins | % of total proteins |
|---|---|---|---|---|---|
| Small (CC) | GO:0070062 | Extracellular exosome | $6.249 \times 10^{-82}$ | 157 | 68 |
| Large (CC) | GO:0005829 | Cytosol | $3.3639 \times 10^{-53}$ | 305 | 35 |
| Both (CC) | GO:0070062 | Extracellular exosome | $3.411 \times 10^{-161}$ | 299 | 74 |
| Small (BP) | GO:0007229 | Integrin-mediated signalling pathway | 0.02272 | / | / |
| Large (BP) | GO:0006977 | DNA damage response, signal transduction by p53 class mediator resulting in cell cycle arrest | $4.730 \times 10^{-06}$ | / | / |
| | GO:0090090 | Negative regulation of canonical Wnt signalling pathway | $1.294 \times 10^{-04}$ | / | / |
| | GO:0090263 | Positive regulation of canonical Wnt signalling pathway | $4.878 \times 10^{-04}$ | / | / |
| | GO:0006987 | Activation of signalling protein activity involved in unfolded protein response | 0.01973 | / | / |
| Both (BP) | GO:0006977 | DNA damage response, signal transduction by p53 class mediator resulting in cell cycle arrest | $2.725 \times 10^{-05}$ | / | / |
| | GO:0048013 | Ephrin receptor signalling pathway | $9.055 \times 10^{-05}$ | / | / |
| | GO: 0090263 | Positive regulation of canonical Wnt signalling pathway | $1.011 \times 10^{-04}$ | / | / |
| | GO:1900740 | Positive regulation of protein insertion into mitochondrial membrane involved in the apoptotic signalling pathway | 0.00223 | / | / |
| | GO:0006987 | Activation of signalling protein activity involved in unfolded protein response | 0.00233 | / | / |
| | GO: 0090090 | Negative regulation of canonical Wnt signalling pathway | 0.00321 | / | / |
| | GO:0048010 | Vascular endothelial growth factor receptor signalling pathway | 0.01213 | / | / |

Extracellular exosome was the most common cellular component for smEVs, cytosol was the most common for lgEVs. Under biological processes, eight overlapping signalling pathways were identified. Wnt signalling featured often.

preparations (Fig. 3f). The most common GO term associated with proteins in each of our smEV preparations was extracellular exosome (GO:0070062; Supplementary Data 1), suggesting, as with the WT smEVs, that our preparations enriched for MVB-associated vesicles. Enrichment analysis using GetGo to identify signalling pathways (under biological processes) identified ten different signalling pathways, six of which were enriched in both $Bbs4^{-/-}$ and $Bbs6^{-/-}$ populations (Table 2). These included both a positive and negative regulation of canonical Wnt signalling. Closer inspection of Wnt signalling proteins in these different EV populations revealed that cilia mutant smEVs contained almost twice as many Wnt signalling proteins compared to control (Table 3). The additional Wnt signalling proteins found in cilia mutant smEVs could be detected in both ($Bbs4^{-/-}$ and $Bbs6^{-/-}$) subpopulations, with only a few exceptions. Up to 50% of these additional proteins were highly enriched, a large proportion of which had only been found in control lgEVs (red lettering) and were absent from control smEVs.

This suggests that loss of cilia function triggers a shift in the mode of release of Wnt signalling proteins. Despite differing molecular mechanisms underlying ciliary dysfunction between the two mutant cells lines, we still observed similar trends, suggesting a universal role for cilia in influencing EV composition and release.

**Ciliary smEVs contain miRNAs targeting Wnt signalling.** smEVs have also been reported to contain various small RNAs. RNA profiling of our smEV pellets revealed enrichment of small RNAs and a lack of cellular 18S and 28S ribosomal RNA as in the corresponding source cells (Supplementary Fig. 4a). In an effort to identify the small RNA cargo, we performed small RNA sequencing from three independent smEV pellets from each cell line. miRNA represented ~48–64% of small RNAs identified in our smEV pellets and was the most abundant biotype identified by far (Supplementary Data Table 4). Principal component analysis showed that the three replicates from each cell line were

more similar to one another than across cell lines, indicating that miRNA expression differences were cell line-specific (Supplementary Fig. 4b).

Comparison of miRNA expression between ciliary mutant and control smEVs showed 24 differentially loaded miRNAs (adjusted $P$ value <0.1 and log2 fold change ≥2 or ≤ −2) out of 235 detected (Fig. 4a–d and Supplementary Data Table 5). smEV pellets derived from $Bbs4^{-/-}$ cells contained 21 differentially loaded miRNAs compared to control EVs (Fig. 4a, c and Supplementary Data 5), nine of which were down- and twelve upregulated. smEV pellets derived from $Bbs6^{-/-}$ cells contained only twelve differentially loaded miRNAs, four were down- and eight upregulated (Fig. 4b, d). All miRNAs upregulated in $Bbs6^{-/-}$ smEVs, were also upregulated in $Bbs4^{-/-}$ smEVs, although only one miRNA (miR-1983) was downregulated in both. We identified potential target genes of differentially loaded miRNAs using two prediction tools (TargetScanMouse and miRDB). Approximately 50% of differentially loaded miRNAs had predicted Wnt-related target genes and are listed in Table 4. Most genes were targeted by more than one miRNA, such as APC, Ctnb1 and Ctnnd2. All Wnt-related miRNAs were upregulated in both ciliary mutant smEVs, downregulated Wnt-related miRNAs were only found in $Bbs4^{-/-}$ sEVs (Fig. 4c, d).

Since a high proportion of differentially loaded miRNAs had Wnt-related target genes and a high number of Wnt-related proteins were selectively secreted via smEVs in cilia mutants, we speculated that cilia associated smEVs might contribute to the regulation of a Wnt response either in secreting cells or in target cells.

**Cilia-related smEVs modulate the cellular Wnt response of possible target cells.** Since many of the differentially loaded molecules (proteins and miRNA) between control and mutant smEVs related to Wnt signalling, we tested whether these might exert a functional effect in vivo. First, we examined whether our isolated smEVs were taken up by target cells. Fluorescently

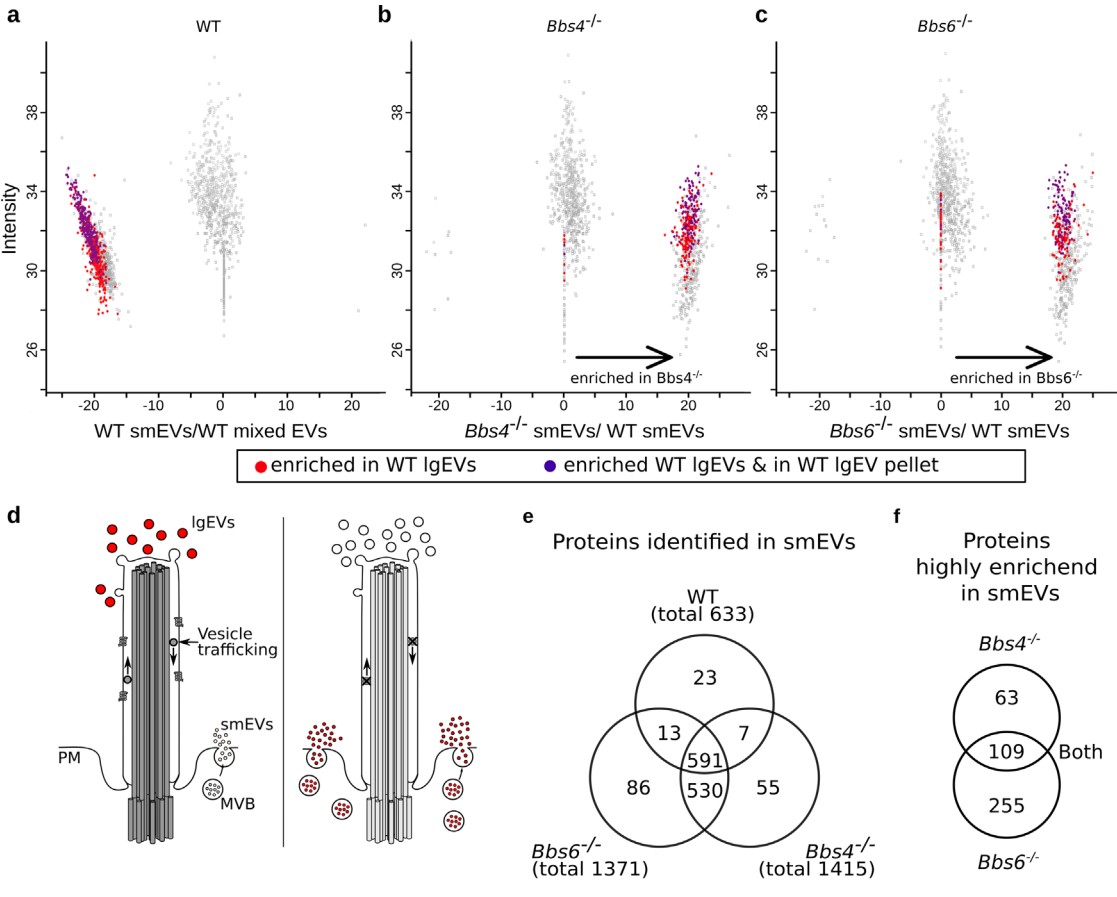

**Fig. 3 Cilia dysfunction alters EV protein composition. a–c** Proteins identified via liquid chromatography-mass spectrometry analysis in four out of six pellets isolated from control and ciliated mutant KM cell supernatant. Source data are provided as a Supplementary data file. **a** Repeat of scatter plot in Fig. 2c depicted to demonstrate change in the mode of protein release from lgEVs in control to smEVs upon cilia dysfunction for the subset of proteins. **b**, **c** Proteins identified in smEV preparations from control vs. cilia mutant cells. Proteins in the right cluster were enriched in smEVs released from mutant cells. Proteins demarked in red and purple were proteins considered enriched in control lgEVs and absent from control smEVs. Their shift to the right cluster suggests an altered method of release from mutant cells. **d** Simplified schematic representation of possible modes of EV release. lgEVs are released at the ciliary tip and smEVs from the cell membrane. Upon ciliary dysfunction, this is hampered and increased MVB-derived smEVs are released directly from the cell membrane (figure created by V.Kretschmer). **e** Absolute numbers of different protein populations depicted in (**b**, **c**). $Bbs4^{-/-}$ and $Bbs6^{-/-}$ smEVs contained twice as many proteins compared to control. **f** Number of proteins significantly enriched ($P$ value $-\log \geq 2$) in $Bbs4^{-/-}$ and $Bbs6^{-/-}$ smEVs compared to the control.

**Table 2 Gene ontology enrichment analysis (getgo.russelllab.org) of proteins highly enriched in ciliary mutant smEVs.**

| EV sample | Term | Signalling pathway | Fisher-C-PV |
|---|---|---|---|
| $Bbs4^{-/-}$ smEVs | GO:0090090 | Negative regulation of canonical Wnt signalling pathway | $6.005 \times 10^{-07}$ |
| | GO:0090263 | Positive regulation of canonical Wnt signalling pathway | $1.099 \times 10^{-06}$ |
| | GO:0048011 | Neurotrophin TRK receptor signalling pathway | 0.002384 |
| | GO:0048013 | Ephrin receptor signalling pathway | 0.00912 |
| | GO:0007179 | Transforming growth factor-beta receptor signalling pathway | 0.01076 |
| | GO:0007173 | Epidermal growth factor receptor signalling pathway | 0.02680 |
| $Bbs6^{-/-}$ smEVs | GO:0090263 | Positive regulation of canonical Wnt signalling pathway | $2.093 \times 10^{-05}$ |
| | GO:0090090 | Negative regulation of canonical Wnt signalling pathway | $7.957 \times 10^{-04}$ |
| | GO:0048013 | Ephrin receptor signalling pathway | 0.00113 |
| | GO:0097193 | Intrinsic apoptotic signalling pathway | 0.00165 |
| | GO:0007229 | Integrin-mediated signalling pathway | 0.00232 |
| | GO:0048011 | Neurotrophin TRK receptor signalling pathway | 0.00881 |
| | GO:0048010 | Vascular endothelial growth factor receptor signalling pathway | 0.02178 |
| | GO:0030512 | Negative regulation of transforming growth factor-beta receptor signalling pathway | 0.04295 |
| | GO:0007173 | Epidermal growth factor receptor signalling pathway | 0.04805 |

All signalling pathways identified under biological processes are listed. In both sets of ciliary mutant smEVs, Wnt-related signalling pathways were the most significantly enriched.

**Table 3 Wnt-related proteins, as determined by UniProt or GetGo (uniport.org; getgo.russelllab.org), found inside control and ciliated mutant EVs.**

| | |
|---|---|
| Wnt proteins in WT lgEVs | Cav1; **Csnk1d**; Csnk2a2; **Ddb1**; G3bp1; **Gsk3b**; **Hdac2**; **Prkaa1**; **Psma1**; **Psma4**; **Psmb3**; Psmb4; Psmb5; **Psmc1**; **Psmc3**; **Psmc4**; **Psmd11**; **Psmd13**; Psmd14; **Psmd3**; Psmd4; **Psmd5**; **Psmd6**; **Psmd7**; Psmd8; **Rab5a**; **Rps27a**; **Snx3**; Stk3; Strn; Uba52; **Vps35** |
| Wnt Proteins in WT smEvs | Cd2ap; Cd44; Cdh1; Cdh2; Csnk2a1; Csnk2b; Ctnna1; Ctnnb1; Ctnnd1; Cxadr; Ddx3x; Egfr; Fermt1; Gnaq; Gnb2l1; Ilk; Itga3; Psma2; Psma2; Psma5; Psma6; Psma7; Psmb1; Psmb6; Psmc2; Psmc5; Psmc6; Psmd1;Psmd2; Rac1; Ruvbl1; Ruvbl2; Slc9a3r1; Src; Vcp |
| Wnt proteins highly enriched in *Bbs4−/−* smEVs | Cpe; **Ddb1**; Fermt2; **Gsk3b**; **Hdac2**; Mark2; **Psma4**; Psmb2; **Psmb5**; **Psmc1**; **Psmc3**; **Psmd11**; Psmd12; **Psmd13**; **Psmd6**; **Psmd7**; Ptpn23; Ptprk; **Rps27a**; Tmem237; Vps4a |
| Wnt proteins present in *Bbs4−/−* smEVs | Atp6ap; Cd2ap; Cd44; Cdh1; Cdh2; Csnk1d; Csnk2a1; Csnk2b; Ctnna1; Ctnnb1M Ctnnd1; Cxadr; Daam1; Ddx3x; Egfr; Fermt1; Gnaq; Gnb2l1; Hgs; Itga3; Macf1; **Prkaa1**; Psen1; Psma1; Psmb3; Psmc4; **Psmd3**; Psmd5; Ptprj; Rab5a; Ruvbl1; Ruvbl2; Sdc1; Skp1; Slc9a3r1; **Snx3**; Src; Vcp; **Vps35** |
| Wnt proteins highly enriched in *Bbs6−/−* smEVs | Cpe; **Csnk1d**; Ctnnd1; Cxadr; Daam1; Fermt2; **Gsk3b**; Hgs; Itga3; Macf1; Psen1; **Psma1**; **Psma4**; Psmb2; **Psmb3**; **Psmc1**; **Psmc3**; **Psmc4**; **Psmd13**; **Pdmd5**; **Psmd6**; **Psmd7**; Ptk7; Ptprk; **Rab5a**; **Rps27a**; Sdc1; Smurf2; Tmem237; Vps4a |
| Wnt proteins present in *Bbs6−/−* smEVs | Cd2ap; Cd44; Cdh1; Cdh2; Csnk2a1; Csnk2b; Ctnna1; Ctnnb1; Ddb1; Ddx3x; Egfr; Fermt1; Gnaq; Gnb2l1; Ilk; Mark2; **Prkaa1**; Psmb5; Psmd11; Psmd12; **Psmd3**; Ptpn23; Ptprj; Ruvbl1; Ruvbl2; Skp1; Slc9a3r1; **Snx3**; Src; Vcp; **Vps35** |

Proteins demarked in bold were originally found in control lgEVs only, but were detected in smEVs upon cilia dysfunction.

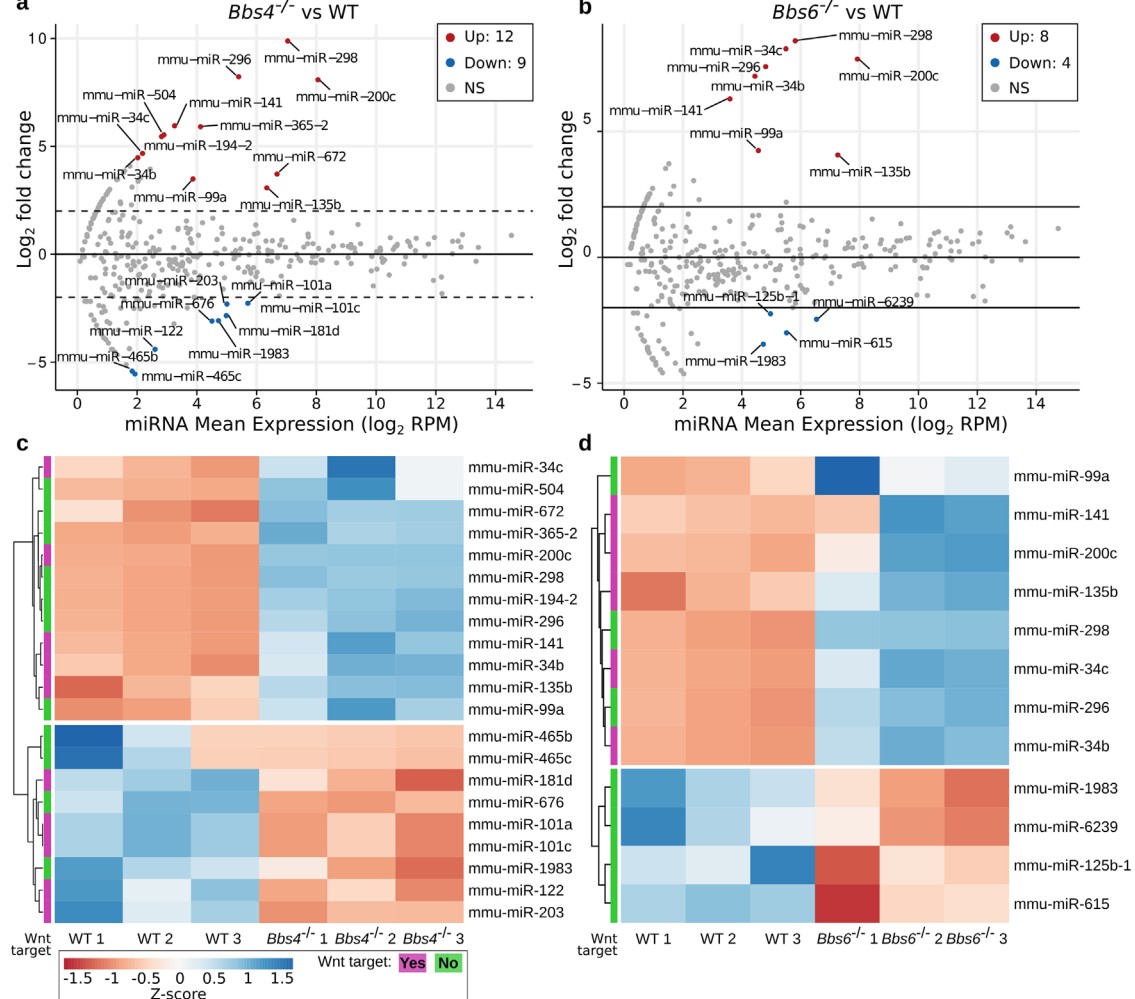

**Fig. 4 smEVs released from cilia mutant cells contain differentially loaded miRNAs involved in Wnt signalling. a**, **b** Volcano blots of miRNAs isolated from three independent smEV prepartions from *Bbs4−/−* (**a**) and *Bbs6−/−* (**b**) cells compared to control. Upregulated miRNAs are depicted by red dots, downregulated by blue dots. **c**, **d** Heatmaps of all differentially loaded miRNAs with a log2 fold change ≥2 or ≤ −2 and adjusted *P* value <0.1 compared to control. miRNAs with target genes related to the Wnt signalling pathway are marked in purple. Source data are provided as a Supplementary data file.

**Table 4 Wnt-related miRNAs and their predicted target genes.**

| miRNA | Possible Wnt-related target genes | | Expression | |
|---|---|---|---|---|
| | TargetScanMouse | miRDB | Bbs4−/− smEVs | Bbs6−/− smEVs |
| mmu-miR-34b | Ctnna2, Ctnna3, Ctnnd1, Dkk2, Fzd1, Fzd2, Fzd5, Sfrp1, Sox17, Wisp1, Wnt2, Wnt2b, Wnt3, Wnt5a, Wnt9b, Wnt10a | Ctnnb1, Ctnnd2, Fzd, Sfrp5, Wnt5a | ↑ | ↑ |
| mmu-miR-34c | Ctnna3, Fzd3, Fzd5, Fzd8, Wif1, Wls, Wisp1, Wnt2, Wnt2b, Wnt4, Wnt7b, Wnt9a | Ctnnd2, Lef1, Wif1, Sfrp5 | ↑ | ↑ |
| mmu-miR-135b | Ctnna1, Ctnna3, Dkk3, Dvl3, Fzd4, Wisp3, Wnt1, Wnt7b, Wnt9a | Wnt3 | ↑ | ↑ |
| mmu-miR-141 | Apc, Ctnna3, Ctnnal1, Ctnnd1, Ctnnd2, Dkk3, Dvl2, Wnt2, Wnt3a, Wnt4, Wnt5a, Wnt5b, Wnt8b, Wnt9b | Apc2, Ctnnb1, Ctnnd2, Dkk3, Wnt5b, Wnt9b | ↑ | ↑ |
| mmu-miR-200c | Ctnna3, Dkk2, Dvl3, Fzd3, Fzd6, Fzd9, Sfrp1, Wnt2b, Wnt4, Wnt1l | Apc, Ctnnd2, Dkk2 | ↑ | ↑ |
| mmu-miR-101a | Apc, Ctnna2, Ctnna3, Ctnnal1, Ctnnbip1, Ctnnd1, Dkk3, Dvl3, Fzd4, Fzd6, Fzd9, Wisp3, Wnt2b, Wnt8a | Apc, Fzd4, Fzd6, Wnt2b | ↓ | / |
| mmu-miR-101c | Apc, Ctnna2, Ctnna3, Ctnnal1, Ctnnd1, Dkk4, Dvl1, Fzd3, Fzd4, Fzd5, Fzd8, Fzd9, Wisp3 | Dkk4, Fzd4 | ↓ | / |
| mmu-miR-122 | Ctnnb1, Wisp1, Wnt4 | Apc, Ctnna1, Ctnnb1, Ctnnd1, Dvl1, Sfrp1, Wnt9a | ↓ | / |
| mmu-miR-181d | Apc, Ctnnd1, Fzd1, Fzd9, Wnt5a, Wnt8a | Dkk1, Wls | ↓ | / |
| mmu-miR-203 | Apc, Ctnna2, Ctnna3, Ctnnal1, Ctnnd1, Ctnnd2, Dvl2, Fzd4, Fzd5, Fzd7, Fzd8, Wls, Wnt2, Wnt4, Wnt5a, Wnt7a, Wnt7b, Wnt1l | Lef1, Wnt7a | ↓ | / |

All significantly up- or downregulated miRNAs with predicted Wnt target genes found in Bbs4−/− and Bbs6−/− smEVs are listed. All upregulated (↑) miRNAs were upregulated in both smEV samples. Five Wnt-related miRNAs were found to be downregulated (↓) in Bbs4−/− smEVs and depleted in Bbs6−/− smEVs. Target prediction was done using TargetScanMouse 7.1 (TargetScanMouse 7.1) and miRDB (miRDB - MicroRNA Target Prediction Database).

labelled smEVs isolated from control and mutant KM cells were applied to HEK293T cells. Confocal microscopy revealed an accumulation of green puncta within cells treated with labelled smEV preparations (Fig. 5a). Since green puncta were detected at the same level as cell nuclei (Z-stack projections at the bottom of the composite), we propose that labelled EVs are indeed taken up by HEK293T cells, and not just attached to the cell surface. Cells treated with preparations made with medium only showed virtually no fluorescent puncta (Fig. 5a, b), suggesting that the observed uptake was not due to particle contamination from the medium. Quantification of smEV uptake revealed that despite loading the same number of particles, smEVs derived from Bbs4−/− and Bbs6−/− KM cells were taken up less frequently than smEVs derived from control. Despite this, these fewer smEVs were still able to elicit a response in target cells.

Using a TCF/LEF Reporter (luc)–HEK293 Cell Line, which contains a stably integrated firefly luciferase gene under the control of TCF/LEF responsive elements, we were able to monitor the activity of the WNT/β-catenin signalling pathway after application of smEVs. An equal number of smEVs were applied to reporter cells for 24 h. SmEVs from control KM cells significantly reduced the baseline luciferase activity by ~30% (Fig. 5c). Although smEVs derived from cilia mutant cells were much less readily taken up, they were still able to significantly dampen the WNT response in target cells. A medium only preparation did not alter luciferase activity, indicating that our medium did not contain contamination of WNT components. WNT3a conditioned media was used as a positive control and serum-starved cells were used as a negative control. To determine that these effects were coming from smEVs, we isolated CD9/CD81/CD63-positive vesicles using an exosome isolation kit (via magnetic beads). CD9/CD81/CD63-positive vesicles were similarly able to dampen WNT/β-catenin activity in our reporter cell (Fig. 5d). This implies that the dampening effect shown in untreated pellets is largely a consequence of CD9/CD81/CD63-positive vesicles. Since our smEV preparations also contained miRNAs, we next sought to determine whether these also influenced this WNT response. To deplete smEVs of miRNAs, we knocked down Dicer in our KM cells prior to harvesting smEVs. The riboendonuclease Dicer is a critical regulator of miRNA biogenesis[61,62]. smEVs from Dicer knockdown cells show the same WNT dampening effect than cells treated with non-targeting control (Fig. 5e). Since Dicer knockdown prior to smEV harvesting made no difference on their ability to dampen the WNT response, we propose that smEV protein content largely drives the responsiveness in this reporter cell line.

To determine whether our observed mechanisms might be reflected in a ciliopathy patient disease model, we generated urine-derived renal epithelial cell (UREC) cultures from a BBS10 patient with age and gender-matched control. TEM micrographs from UREC supernatant revealed similar spherical biconcave structures 100–150 nm in size (Fig. 5f). The concentration and size distribution of smEVs isolated from URECs was measured via NTA (Fig. 5g), which similarly revealed a dramatic increase in vesicle number in patient samples. Although UREC smEVs from the control was not able to affect the WNT response in our TCF/LEF Reporter cell line, UREC derived smEVs from the patient did dampen WNT/β-catenin activity (Fig. 5h).

**Selective secretion of WNT components may influence WNT signalling in cilia mutant source cells.** Secretion of WNT molecules has recently been found to also affect source cells[63,64]. Since a large percentage of differentially loaded proteins selectively secreted via cilia mutant cells were proteasomal subunits (including Psma1, Psma4, Psmb3, Psmb5, Psmc1, Psmc3, Psmd3,

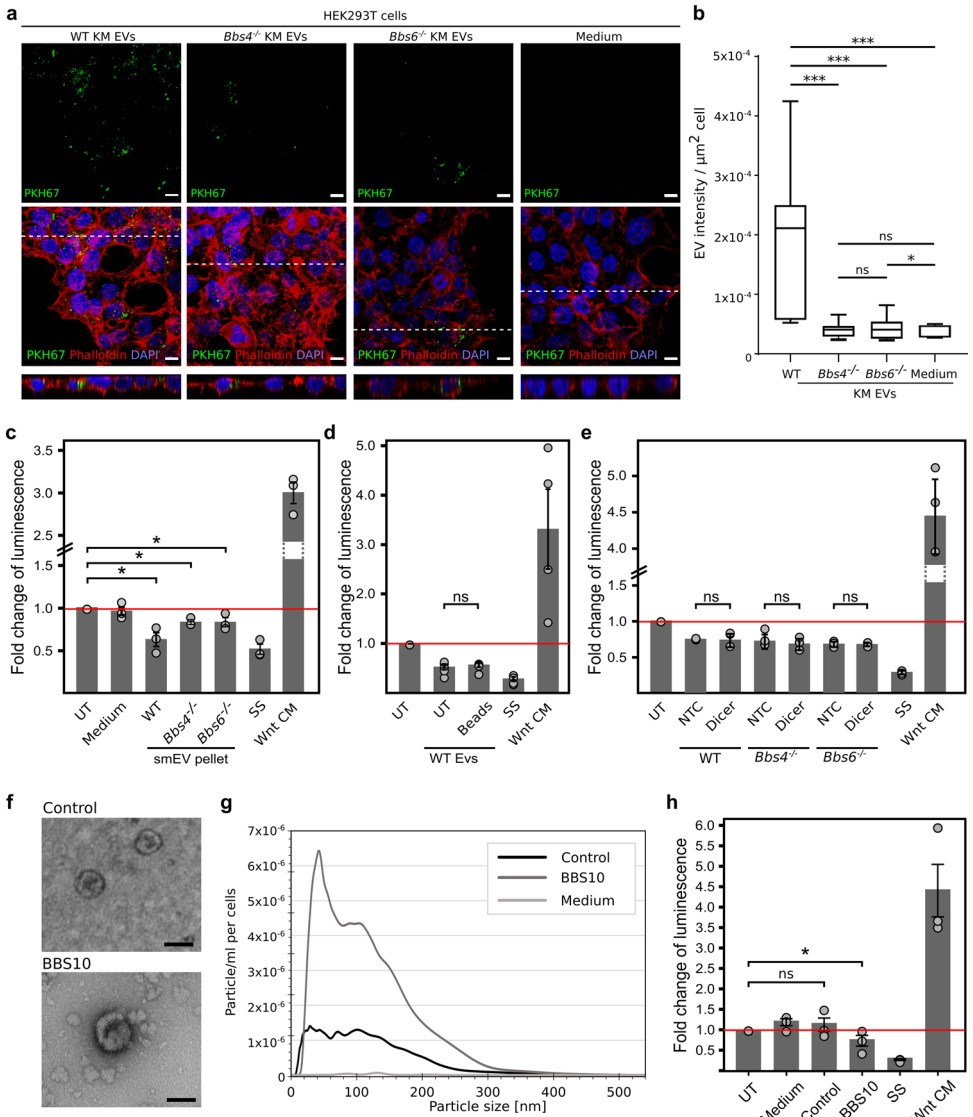

**Fig. 5 Small EVs are bioactive and modulate the cellular Wnt response. a** HEK293T cells (Phalloidin (cell membrane), DAPI (nuclei)) were able to take up fluorescently labelled EVs (green dots, PKH67 staining) harvested from ciliated control and mutant KM supernatant. Medium controls show low background staining (scale bar = 25 μm; figure shows representative images from three independent experiments). **b** Quantification of fluorescently labelled EV signal per μm² as shown in (**a**). A significantly higher fluorescent signal was detected in HEK293T cells incubated with smEVs derived from control KM cells compared to cilia mutant cells (ₙWT, *Bbs4*⁻/⁻, *Bbs6*⁻/⁻ EVs, medium = 13 analysed images from three independent experiments; two-sided Mann–Whitney *U* test pWT EV vs. *Bbs4*⁻/⁻ EV = 0.00013; pWT EV vs. *Bbs6*⁻/⁻ EV = 0.000072; pWT EV vs. medium = 1.923E-7; p*Bbs4*⁻/⁻ EV vs. medium = 0.064; p*Bbs6*⁻/⁻ EV vs. medium = 0.05). **c** Purification of smEVs and application on TCF/LEF reporter-HEK293T cells. Relative Wnt activity of smEV-treated TCF/LEF HEK cells as measured via luminescence. Vesicle number was measured via NTA. The applied vesicle number was normalised to control EV number. Application of smEVs derived from all ciliated KM cells significantly decreased the Wnt response of target cells (ₙWT, *Bbs4*⁻/⁻, *Bbs6*⁻/⁻ = four independent experiments; two-sided Mann–Whitney *U* test pWT vs. TCF = 0.028; p*Bbs4*⁻/⁻ vs. TCF = 0.028; p*Bbs6*⁻/⁻ vs. TCF = 0.047) with no significant difference between the smEVs derived from control or ciliary mutant samples (two-sided Mann–Whitney *U* test pWT vs. *Bbs4*⁻/⁻ = 0.713; pWT vs. *Bbs6*⁻/⁻ = 0.551). Medium control shows no difference in Wnt readout (ₙMedium = three independent experiments). WNT3a conditioned media was used as a positive control and serum-starved (ss) cells were used as a negative control. **d** Luciferase assay with control KM EVs purified via differential centrifugation (UT) and magnetic beads (Beads) showed no difference in the ability to dampen Wnt signalling in target cells (ₙWT EV UT, WT EV Beads = five independent experiments; two-sided Mann–Whitney *U* test pWT EV UT vs. WT EV Beads = 0.690. **e** Dicer siRNA-mediated knockdown inside control and mutant KM cells prior EV purification showed no difference in capability to dampen Wnt signalling inside target cells. (ₙDicer NTC, Dicer kd = three independent experiments; two-sided Mann–Whitney *U* test pWT NTC vs WT Dicer = 0.7; unpaired *t* test p*Bbs4*⁻/⁻ NTC vs. *Bbs4*⁻/⁻ Dicer=0.775; p*Bbs6*⁻/⁻ NTC vs. *Bbs6*⁻/⁻ Dicer = 0.893). **f** Morphological properties of isolated UREC EVs were determined via negative staining followed by transmission electron microscopy and revealed 100–150 nm vesicles (scale bars: 100 nm, experiment was done two times). **g** Nano Particle Tracking Analysis (NTA) measurements from control UREC cells show less smEV release compared to BBS10 cells. Medium control contained no particles (ₙControl, BBS10, medium = three independent experiments). **h** Luciferase assay of isolated UREC EVs showed reduced Wnt signalling in target cells upon application of BBS10 EVs (ₙBBS10 = three independent experiments). EVs from control cells showed no effect (ₙControl = three independent experiments; unpaired *t* test pUT vs. BBS10 EV = 0.05; pUT vs. control EV = 0.385). **b** Boxplots show median, interquartile range, and maximum and minimum within 1.5 interquartile range. **c–e**, **h** Data are presented as mean values +/− SEM. n.s., *P* > 0.05, *P ≤ 0.05, **P < 0.01, ***P < 0.001.

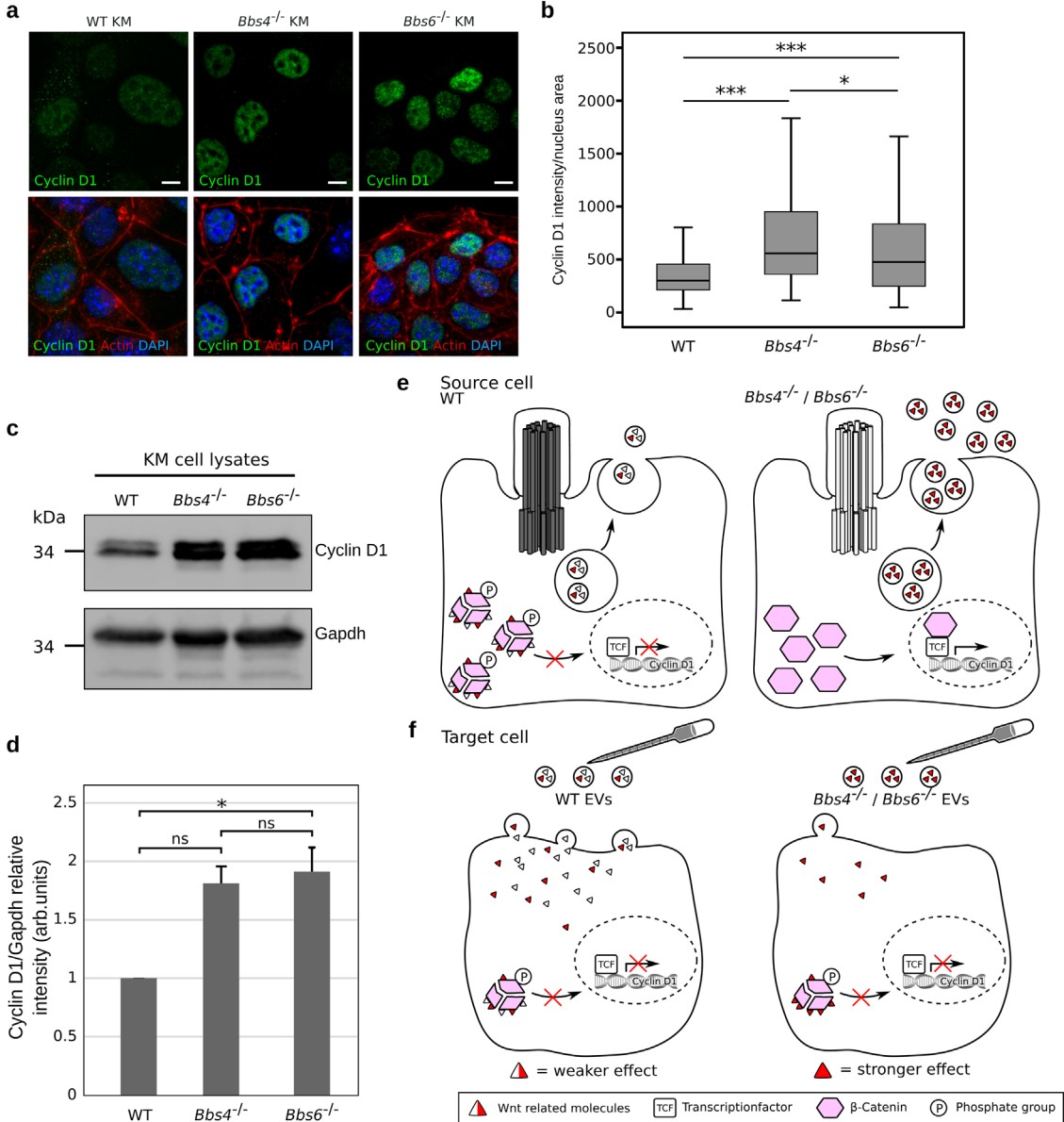

**Fig. 6 Control and ciliary mutant cells show differences in Wnt signalling. a** Representative immunofluorescent images of nuclear Cyclin D1 (green) staining in control and ciliary mutant KM cells (Phalloidin (cell membrane), DAPI (nuclei)). **b** Quantification of Cyclin D1 intensity inside the nucleus showed significantly stronger intensities inside the ciliary mutant cells, indicating an upregulation of Wnt signalling ($_NWT = 389$ cells; $_NBbs4^{-/-} = 330$ cells, $_NBbs6^{-/-} = 428$ cells; two-sided Mann–Whitney $U$ test pWT vs. $Bbs4^{-/-} = 4.19E-27$; pWT vs. $Bbs6^{-/-} = 1.09E-15$; p$Bbs4^{-/-}$ vs. $Bbs6^{-/-} = 0.02$). **c** Representative western blot image of Cyclin D1 in control and ciliary mutant KM cells. The experiment was done four times. **d** Quantification of Cyclin D1 relative band intensity identified a significant difference between control and $Bbs6^{-/-}$ KM cells ($N = 4$; unpaired $t$ test pWT vs. $Bbs6^{-/-} = 0.046$, Data are presented as mean values $+/-$ SEM), but no difference between both ciliary mutants or WT and $Bbs4^{-/-}$ cells ($N = 4$; unpaired $t$ test p$Bbs4^{-/-}$ vs. $Bbs6^{-/-} = 0.767$; pWT vs. $Bbs6^{-/-} = 0.056$, Data are presented as mean values $+/-$ SEM). **e, f** Possible schematic representation of EV shedding from control and ciliary mutant cells and its effect on target cells. **e** Ciliary mutant cells release more MVB-derived EVs with different content from the plasma membrane. Mutant cells exhibit higher levels of WNT signalling (figure created by V.Kretschmer). **f** Application of the same number of EVs on target cells leads to a Wnt dampening effect. Since less $Bbs4^{-/-}$ and $Bbs6^{-/-}$ EVs are taken up than control EVs, it suggests that an individual $Bbs4^{-/-}$ or $Bbs6^{-/-}$ EV exerts a stronger Wnt dampening effect (figure created by V.Kretschmer). **b** Boxplots show median, interquartile range, and maximum and minimum within 1.5 interquartile range. n.s., $P > 0.05$, *$P \leq 0.05$, **$P < 0.01$, ***$P < 0.001$.

Psmd5, Psmd6, Psmd7, Psmd11, Psmd13), we speculated that this might affect proteasomal degradation of β-catenin in source cells. Numerous papers have already shown that suppression of BBS proteins leads to stabilisation of β-catenin and perturbed proteasomal degradation[37,42,46,65]. Active β-catenin translocates into the nucleus and activates transcription of TCF target genes such as *Cyclin D1*. In line with this, we observed that control KM source cells had lower levels of Cyclin D1 expression both on immunocytochemistry and on western blot (Fig. 6a–d). Elevated Cyclin D1 expression in both cilia mutant cell lines is consistent with lower levels of proteasomal degradation of β-catenin.

## Discussion

The primary cilium plays a central role in signal transduction and cell communication. Although it has been shown that cilia are able to release EVs[8], few studies have been performed in

mammalian tissues, nor is it known whether these EVs are biologically active. Further, most studies examining cilia associated EVs have focused on lgEVs budding directly from the plasma membrane. In light of this, we set out to characterise the smEV phenotype in ciliary mutant mammalian cells generated from a disease-relevant tissue and determine how these may participate in signalling mechanisms.

Using immortalised mouse kidney medullary cell lines in which ciliary trafficking is disrupted ($Bbs4^{-/-}$ and $Bbs6^{-/-}$), we found that mutant cells released almost four times as many small vesicles compared to control. We could also recapitulate this finding in UREC cultures derived from a *BBS10* patient. These data complement the work done in *C. elegans* showing that BBSome proteins negatively regulate EV shedding[11]. The origin of these additional vesicles might elucidate novel cilia functionalities.

Studies in different species have shown variations in ciliary EV shedding[8], some describe vesicles budding from the axonemal membrane[66,67], yet the most common site is at the ciliary tip[18,66] particularly in mammalian cells[22,24,29–31]. Vesicles released from the ciliary tip are most likely lgEVs formed via constriction of the ciliary membrane[9,21,22]. Numerous studies have shown the requirement of intraflagellar transport (IFT) along the ciliary axoneme in the biogenesis of these larger vesicles[11,19,26,27,68]. A previous study showed an increase in ciliary tip decapitation in cilia mutants[22]. We concentrated our efforts on characterising smEVs. Our finding that inhibition of MVB formation via GW4869 treatment or Alix knockdown significantly reduces small-vesicle release, suggests that a large proportion of excess smEVs are actively secreted via an MVB-associated mechanism. This hypothesis is strengthened by the observation that Annexin knockdown, important for the biogenesis of lgEVs, did not reduce small-vesicle release.

Several reports in various organisms and cell lines have already implicated BBSome proteins in endocytic functions at the base of the cilium[69–71]. The different responses between our mutant cell lines might reflect differences in protein function, as well as their association with different mechanisms of MVB biogenesis. GW4869 inhibits neutral sphingomyelinases (nSMase) and thereby ESCRT-independent MVB formation, whereas Alix knockdown influences the ESCRT-dependent pathway[58,59,72–78]. Both mechanisms appear to be involved in contributing to additional smEV release upon loss of *Bbs4*, yet only ESCRT-dependent MVB release was increased upon loss of *Bbs6*.

It is unclear where these excess MVB-associated EVs are coming from. They are less likely to be coming from the ciliary tip since MVBs with a diameter of 400–1000 nm[79] are too large to be transported into the cilium, nor has an MVB ever been observed in the ciliary axoneme. A potential source of smEV release via MVBs is the ciliary pocket, a subdomain of the plasma membrane surrounding the base of the cilium[80]. Although it has been shown that *C. elegans* shed EVs from the ciliary base[12,25], so far no MVB-derived structures have been reported at this region. This might reflect the unique structure of these highly specialised ciliated neurons. In *Trypanosomes*, EVs are released from the flagellar membrane as well as the flagellar pocket (akin to the ciliary pocket in mammalian cells[81,82]). The flagellar pocket is a unique site for endo- and exocytosis (reviewed in refs. [83,84]) and although *Trypanosomes* do not have MVBs, they do have late endosomes containing orthologues of the ESCRT machinery commonly found at the base of the cilium[85]. Intriguingly, in support of our hypothesis, in *T. brucei* BBSome components have been localised to membrane and vesicles at the flagellar pocket and BBSome loss perturbs endocytic trafficking at this region[69]. In metazoans, however, the total endocytic capacity of the ciliary

pocket remains minor relative to the rest of the cell surface[86]. Although the ciliary pocket might not be the preferential site for endo- and exocytosis in functional cilia[80], this might change if ciliary trafficking is disrupted and shedding at the tip is obstructed. In support of this, the accumulation of secretory granules around the ciliary pocket[87,88] and the presence of tubular structures[89,90] suggest that the ciliary pocket could serve as a site for docking and fusion for secretory vesicles.

To identify a possible biological function of EV release, we examined EV content in more detail with a particular focus on the content of smEVs. In control cells, proteins found in smEVs contributed to similar pathways as proteins in lgEvs and included proteins related to responses to cellular stress; including DNA damage response (GO:0006977), unfolded protein response (GO:0006987) and apoptotic signalling response (GO:1900740). We also found highly enriched proteins involved in developmental processes including Wnt signalling (GO:0090090, GO:0090263), Eph receptor signalling (GO:0048013), Vegf receptor signalling (GO:0048010) and integrin-mediated signalling pathways (GO:0007229).

In control cells, we identified a distinct set of proteins more readily loaded into lgEVs. When comparing these to the content of ciliary mutant smEVs, we noticed that a high percentage were loaded into smEVs upon disruption of cilia function (compare Fig. 3a–c). Thus, suggesting that their original loading was influenced by ciliary processes.

A closer look at the differentially loaded molecules found in ciliary mutant smEVs identified components related to signalling pathways. These include signalling pathways already known to be associated with the cilium such as Wnt signalling and TGF-β signalling[1], as well as others such as neurotrophin TRK receptor signalling, Eph receptor signalling and EGF receptor signalling. The most significantly enriched signalling pathways associated with ciliary mutant smEVs where both positive as well as negative regulation of Wnt signalling. smEV released from cilia mutant cells had almost twice as many Wnt-related proteins. We noted that many of the differentially loaded Wnt signalling proteins are not just involved in Wnt signalling but are also more general regulators of cellular processes. These include the Psm protein subunits which are involved in general protein degradation, Snx3 and Vps35 which are subunits of the retromer involved in retrograde trafficking[91,92] and Prkaa1 (AMP-kinase)[93] and Gsk3beta which are general kinases under the control of various signalling pathways. We also performed small RNA sequencing of smEV content and found miRNAs to be the most abundant small RNA biotype present. Although we only found 21 (in *Bbs4*) and 12 (in *Bbs6*) differentially loaded miRNAs in mutant-derived smEVs, over half of the differentially loaded miRNAs had downstream Wnt signalling targets.

In addition to characterising a population of mammalian smEVs, we were also able to show that they are biologically active. Due to the presence of Wnt-related cargo, we tested whether isolated smEVs would be taken up by target cells and initiate a response, although we are very aware that many of the differentially loaded Wnt signalling proteins are not just involved in Wnt signalling, but might also modify other signalling pathways. Finding an appropriate target cell is challenging since it is not clear to what extent smEVs travel in vivo. Because our source cells came from kidney medullar tissue, we chose another mammalian kidney cell line as a target. We could show that smEVs isolated from control KM cells were readily taken up by HEK293T cells and could actively dampen the Wnt response in TCF/LEF Reporter (luc)–HEK293 cells. Despite ciliary mutant cells releasing more smEVs, these ciliary mutant-derived smEVs were

less able to be taken up by target cells. Nonetheless, they were still equally able to significantly dampen the WNT response in target cells. If we take into account that under physiological conditions ciliary mutant cells release many more EVs, the dampening effect inside target cells would be stronger.

Despite the presence of miRNAs with Wnt-related target genes, we propose that smEV protein content largely drives the responsiveness in this reporter cell line since manipulation of miRNA biogenesis in KM cells prior to EV harvesting did not alter responsiveness. Two seminal papers showed that active WNT ligands can be secreted on the surface of smEVs and are able to induce WNT signalling activity in target cells[55,94]. The most common WNT ligands detected in mammalian EVs thus far are WNT3a, WNT5[55] and WNT11[95–97], although none of these was identified in our study. A high percentage of smEV proteins were proteasomal subunits which could possibly contribute to proteasomal degradation of β-catenin in target cells. smEVs derived from cilia mutant cells had significantly more proteasomal subunits. Similarly, glycogen synthase kinase 3 (GSK3), one of the key regulators of WNT/β-catenin signalling[98] as part of the cytoplasmic destruction complex which regulates the phosphorylation and proteasomal degradation of β-catenin[98], was highly enriched in smEVs derived from cilia mutant cells. The increased abundance of such molecules in mutant-derived EVs might explain the same level of responsiveness on target cells, despite less uptake (Fig. 6f).

An alternative method of smEV determined WNT manipulation may be the selective secretion of downstream WNT components which may influence the Wnt response in secreting cells. Numerous papers have shown that suppression of BBS proteins leads to a stabilisation of β-catenin and perturbed proteasomal degradation[37,42,46,65]. Since β-catenin is a complex molecule and activation or inactivation is dependent on various post-translational modifications, we chose to confirm increased activation of canonical Wnt signalling in our cilia mutant KM cells by examining the expression of Cyclin D1, a downstream transcription factor of β-catenin. The tetraspanins CD9 and CD82 have already been shown to downregulate canonical WNT signalling via assisting smEV discharge of ß-catenin, independent of proteasomal degradation in HEK293T cells[63]. We identified Cd9 and Cd82 in smEV preparations from all three cell lines, with a slightly higher abundance in mutant-derived EVs. Similarly, it was recently shown that upon active WNT/β-catenin signalling up to 70% of cellular GSK3 can be packaged into multivesicular bodies (MVBs), leading to a stabilisation of β-catenin in the source cell[64,99–101]. The exclusive detection of Gsk3β in smEVs derived from cilia mutant cells could explain why cilia mutant source cells have higher levels of Wnt activity due to Gsk3β expulsion. We also detected the ESCRT-0 protein hepatocyte growth factor-regulated tyrosine kinase substrate (Hgs) as well as the v-SNARE receptor synaptobrevin homologue YKT6 (Ykt6) exclusively inside ciliary smEVs. Both of which play a role in sorting WNT signalling proteins for exosomal secretion[55,102]. It is unclear whether this phenomenon is causal or reactionary, but it does suggest that the delicate balance of Wnt homoeostasis is disrupted upon ciliary dysfunction. It was also intriguing that pathways analyses identified both positive and negative regulators of Wnt signalling. This might reflect the ability of smEV release to influence not only target cells but also secreting cells.

These observations lead us to propose a model in which ciliary mutant cells exhibit increased levels of canonical Wnt signalling concomitant with increased release of smEVs loaded with significantly more Wnt-related molecules (Fig. 6e). Although target cells were less able to take up these EVs, their content was still able to initiate a Wnt response (Fig. 6f).

Although the release of lgEVs is gaining a lot of attention in the ciliary field, the association of cilia and smEVs (in particular exosomes) has not been examined. We showed that cilia dysfunction leads to increased smEV release and altered protein sorting into EVs. The abundance of Wnt signalling molecules differentially sorted in mutant EVs suggests smEV-dependent ciliary signalling mechanisms might contribute to the defective Wnt phenotype in numerous cilia mutant cell and animal models[35–42]. In support of this, we were pleased to see that smEVs derived from ciliopathy patient renal tissue significantly dampened the Wnt response in target cells although smEVs from control tissues did not. Therefore our findings could provide insights into ciliopathy disease pathogenesis and intercellular communication.

## Methods

**Cell culture and treatments.** All cells were grown at 37 °C with 5% $CO_2$ in DMEM (Thermo Fisher Scientific) supplemented with 10% FBS and 1% penicillin–streptomycin unless otherwise stated. Wild-type and knockout ($Bbs6^{-/-}$ and $Bbs4^{-/-}$) KM cells[50] were cultivated in DMEM/F-12 (Thermo Fisher Scientific) and serum-starved for 24 h to induce ciliogenesis. TCF/LEF Reporter – HEK293 cells (#60531, BPS Biosciences, Inc.) were grown in minimum essential medium (MEM, Thermo Fisher Scientific) supplemented with 10% FBS, 1% penicillin–streptomycin, 1% non-essential amino acids (NEAA, Thermo Fisher Scientific) and 1 mM Na pyruvate (Carl Roth). HEK293T cells (ATCC®) grown under regular conditions were used for EV uptake. URECs were isolated from the urine of a BBS10 patient (C91fs95X, Exon 2;) and age- and gender-matched healthy control subject. Urine was collected with informed consent and ethical approval (Landesärztekammer Rheinland-Pfalz 2019-14118). URECs were cultivated as described in Ajzenberg et al.[103]. Twenty-four hours prior, EV purification medium was changed to serum-free medium.

Confluent KM cells were treated with 10 μM GW4869 (D1629; Sigma-Aldrich) dissolved in DMSO during serum starvation (24 h) prior to EV preparation and NTA analysis. Alix and Annexin1 siRNA-mediated knockdown (mm.Ri.Pdcd6ip.13, mmRi.Anxa1.13, IDT) was performed using Lipofectamine RNAiMax transfection reagent (13778075, Thermo Fisher) using a reverse transfection protocol according to the manufacturer's instructions. Sequences are provided as a Supplementary Table in Supplementary Information. Twenty-four hours post-transfection, cells were serum-starved prior to EV preparation and NTA analysis. The total cell RNA was isolated using TRIzol reagent (15596026; Thermo Fisher Scientific) and qRT-PCR was performed using SYBR-Green reagent (Life Technologies) on a STEP One Plus Real-Time PCR machine (Applied Biosystems; Thermo Fisher Scientific) to elucidate knockdown efficiency.

**Preparation of extracellular vesicles (EVs).** Characterisation and reporting of extracellular vesicle data were done following the MISEV2018 guidelines[15]. EVs were prepared via differential centrifugation. Serum-starved cell culture medium was centrifuged at 1000×g for 10 min to remove cell debris. The supernatant was further centrifuged at 10.000×g for 30 min at 4 °C for large EVs (10 K pellet). For isolation of small EVs, this supernatant was ultracentrifuged at 100.000×g for 2 h at 4 °C using an Optima L-90K ultracentrifuge (100 K pellet; Beckmann Coulter) with swing bucket SW-28 rotor (Beckmann Coulter). Pellets were suspended in Laemmli buffer, particle-free PBS or fresh DMEM/F-12 medium for further analysis. For detection of EVs under serum-fed conditions, cells were grown 24 h in medium with EV-depleted FBS (Thermo Fisher Scientific) prior to purification.

EV isolation using magnetic beads: 100 K pellets were resuspended in 400 μl DMEM/F-12 and incubated for 1 h at RT with CD9/CD81/CD63-positive magnetic beads (Exosome Isolation Kit Pan, mouse, Miltenyi Biotec). Afterwards, vesicles were isolated according to manufactures instructions. In all, 10% FBS was added to the sample before application onto the TCF/LEF target cells.

**Nanoparticle tracking analysis (NTA).** Particles were analysed using the Nanosight LM10 system (Malvern Panalytical, Herrenberg, Germany) equipped with a 532 nm laser and a syringe pump. Data were analysed using the Nanosight 2.3 software (Malvern Panalytical, Herrenberg, Germany). Particle movement was recorded in five videos, 30 s long, at a steady temperature of 23 °C. The average size and number of particles were calculated with regard to donor cell number.

**Transmission electron microscopy.** Isolated smEV pellets were diluted in particle-free PBS with protease inhibitor (Halt™ Protease and Phosphatase Inhibitor Cocktail (100X), 78440, Thermo Fisher Scientific). Samples were applied to carbon-coated copper grids (400 mesh, Electron Microscopy Sciences) and fixed with 1% glutaraldehyde for 5 min. After washing four times in $dH_2O$, grids were treated with 2% uranyl acetate for 1 min and allowed to dry. A FEI Tecnai G2 12 BioTwin

transmission electron microscope (FEI Company, Hillsboro, USA) was used for imaging.

**Western blot analysis**. For western blot analysis, EV pellets were resuspended in 2× Laemmli buffer and whole-cell lysates in 6× Laemmli (6×: 12% SDS, 60% glycerol, 0.012% bromophenol blue, 0.375 M Tris-HCl), run on a 10% SDS-PAGE and transferred to a PVDF membrane. Membranes were blocked with blocking buffer (0,2% Applichem blocking reagent, 10 mM Tris, 150 mM NaCl, 0.04% NaN$_3$) and incubated with primary (overnight, 4 °C) and secondary antibody (1 h). The following antibodies were used: CD9 (1:1000, clone KMC8, BD Pharmingen™), CD81 (1:1000, clone B-11, Santa Cruz), Flotillin-1 (1:1000, #F1180, Sigma-Aldrich), Tsg101 (1:1000, clone 4A10, GeneTex Irvine), Cyclin D1 (1:1000, #55506, Cell Signaling). Secondary antibodies used were IRDye680 and IRDye800 (Rb and Mm 1:10.000; LI-COR Biosciences; rat¸1:10.000; Thermo Fisher Scientific). Fluorescence detection was carried out using the Odessey Fc Imaging System (LI-COR Biosciences).

**Luciferase assay (WNT activity)**. In total, $2 \times 10^5$ TCF/LEF Reporter – HEK293 cells were seeded in a 96-well plate. After 48 h, cells were treated with smEV pellets isolated from KM cells suspended in a complete minimum essential medium (41090036, Thermo Fisher Scientific). EV numbers were normalised to control EV numbers using NTA. Luminescence was measured after 24 h using a Tecan Infinite® M200 Pro plate reader (Tecan Trading AG) using the Dual-Glo® luciferase assay system (Promega) according to the manufacturer's instructions. Cells incubated with Wnt3a conditioned medium served as a positive control, serum-starved cells as a negative control. Dicer1 siRNA-mediated knockdown (mm.Ri.Dicer1.13) was performed as described above. Sequences are provided as a Supplementary Table in Supplementary Information. Twenty-four hours after knockdown, the medium was changed to serum-free medium. After 24 h incubation, EVs were purified and luciferase assay performed.

**Small RNA sequencing**. Small RNAs were isolated from control and mutant smEV pellets in triplicate using the Total Exosome RNA and Protein Isolation Kit (4478545, Invitrogen) according to the manufacturer's instructions. NGS library prep was performed with NEXTflex Small RNA-Seq Kit V3 following Step A to Step G of Bioo Scientific's standard protocol (V16.06) and sequenced on a NextSeq 500/550 High-throughput Flowcell. Read filtering was done by: (1) adapter trimming; (2) removal of reads with low-quality calls; (3) removal of PCR duplicates using the unique molecule identifiers (UMIs); (4) trimming of the UMI sequences and finally (5) removal of reads shorter than 15 nucleotides. miRNA abundance was estimated with mirDeepMApper v.2.0.0.8 (mapper.pl -e -p -h -m -I -j followed by miRDeep2.pl -t mouse -c -d -v) after which a table of miRNA read counts was prepared using only the maximal count for miRNAs with more than one entry. Differential expression analysis was performed with DESeq2.

**EV protein identification via mass spectrometry**. Pellets were prepared containing smEVs ($n = 6$/cell line), lgEVs ($n = 4$/cell line) and mixed EVs ($n = 6$/cell line) via differential centrifugation as depicted in Fig. 1a. For mixed EV pellets, samples were centrifuged at 1000×g for 10 min and at 100.000×g for 2 h at 4 °C. Less ciliated EV samples were taken from control and cilia mutant cells incubated with EV-depleted FBS ($n = 3$/cell line). All pellets were diluted in 6 M Urea–Tris buffer and subjected to liquid chromatography-mass spectrometry. EV eluates were subjected to methanol–chloroform precipitations followed by tryptic cleavage before they were analysed by LC-MS/MS as described in Supplementary Methods. Identification and quantification were performed with MaxQuant (https://maxquant.net/; version 1.6.1.09) against the mouse subset of the Swissprot database (2019_08, #17,027 entries), statistical analysis was done with Perseus as described in supplementary methods (version 1.6.2.3; https://maxquant.net/perseus/). Mass spectrometry data was often represented as a Scatter plot in which the $y$ axis represents the intensity, which is the sum of all raw intensity measurements for each protein. This was only used for illustration purposes. Ratios are calculated from LFQ intensities.

**EV staining and uptake assay**. smEV pellets were stained with 8 µM Pkh67 (Mini67; Sigma-Aldrich) for 5 min. 1% BSA was added for 1 min to bind excess dye. DPBS was added to a final volume of 1 ml and centrifuged at 100.000×g for 70 min at 4 °C. The labelled pellet was resuspended in a complete cell culture medium and applied to HEK293T cells. Medium control contained Pkh67 dye without EVs. After 24 h, cells were fixed with 4% PFA for 10 min and prepared for immunofluorescence imaging. Cells were permeabilized for 15 min with PBSTX, blocked for 1 h with FishBlock and incubated 1 h with TRITC-Phalloidin (1:200). Cells were imaged on a Leica TCS SP5 II microscope (Leica, Bensheim, Germany) with a 405 Diode, Argon and DPSS 561 laser. Images were processed and quantified using Fiji/ImageJ 64 v5 software (NIH, Bethesda, USA). EV intensity was measured per cell area and background fluorescence was subtracted from individual images.

**Immunofluorescence of KM cells and cilia**. KM cells were incubated for 24 h with serum-free or EV-depleted FBS prior fixation with 4% PFA for 15 min. Cells were washed three times with PBS, before formaldehyde quenching with NH4Cl for 10 min at RT. Cells were permeabilized with 0.3% PBS-Triton X for 15 min followed by blocking with fish block Triton X 0.3% for 1 h at RT. Cell were incubated with primary antibodies overnight at 4 °C (Arl13b, 1:800, #17711-1-AP, Proteintech; GT335, 1:200, #AG-20B-0020-C100, Adipogen Life Sciences; Cyclin D1, 1:200, #55506, Cell Signaling). Cells were washed three times with PBS before incubation with appropriate fluorescent conjugated secondary antibodies (anti-mouse IgG Alexa Four 555, 1:400, Thermo Fisher Scientific; anti-rabbit IgG Alexa Fluor 488, 1:400, Thermo Fisher Scientific), counterstained with DAPI (1:8000, #6843, Carl Roth) and TRITC-Phalloidin (1:400, Thermo Fisher Scientific). The cell was imaged on a Leica CTR6000 confocal microscope, with a DM6000 B laser, the monochrome digital camera DFC360 FX and Leica image Software BlindDeblur Algorithm, one iteration step. All images were processed in Fiji ImageJ 64 v5 software.

**Statistical analysis**. All statistical tests were done in SPSS Statistics 23.0 (IBM, USA). Gaussian distribution was tested using Shapiro–Wilk and Kolmogorov–Smirnov test. For Gaussian distributed data, the $t$ test and for non-Gaussian distributed data, the Mann–Whitney $U$ test was used. Statistical differences were considered when $P \leq 0.05$ using the $t$ test. The null hypothesis of the Mann–Whitney $U$ test was rejected at $P < 0.05$.

**Reporting summary**. Further information on research design is available in the Nature Research Reporting Summary linked to this article.

## Data availability

The deep-sequencing data generated in this study have been deposited in the NCBI GEO database under accession code GEO: GSE153227. The mass spectrometry proteomics data generated in this study have been deposited in the ProteomeXchange Consortium database via the PRIDE partner repository under the accession code PXD020466. Characterisation and reporting of extracellular vesicle data were done following the MISEV2018 guidelines[15]. Source data are provided with this paper.

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

## Acknowledgements
The authors would like to thank Stefanie Kornelia Becker and Dominik Reichert for their technical assistance as well as all the members of the BBS patient community for the continued support of this research. The authors would also like to thank Joana Scheiba and Carsten Geiß for help with FACS. This work was funded by the Alexander von Humbodlt Foundation (Sofja Kovalevskaja Award 2014), RLP-Impulse Funds, Inner-universitäre Forschungsförderung (Stufe 1) of the Johannes Gutenberg-University of Mainz and the Deutsche Forschungsgemeinschaft (DFG) SPP 2127 (MA 6139/3-1).

## Author contributions
H.L.M-S., A.-K.V. and E.-M.K.-A. designed the study; A.-K. V., A.F., A.M.d.J.D. and K.B. acquired the data; A.M.d.J.D. and R.F.K. analysed and interpreted the RNA sequencing data; K.B. and M.U. analysed and interpreted the proteomic data; A.-K.V. and V.K. analysed and interpreted the remaining data. A.F. generated and performed all experiments with UREC cells; H.L.M.-S., A.-K.V. and V.K. wrote and/or revised the manuscript.

## Funding

## Competing interests
António Domingues is currently an employee of Dewpoint Therapeutics. The remaining authors declare no competing interests.
