## [Peer Review File · Nature Communications]

REVIEWER COMMENTS

Reviewer #1 (Remarks to the Author):

The paper "Bardet-Biedl Syndrome-associated extracellular vesicles modulate WNT signaling" from Volz and colleagues describes the interesting finding that ciliary mutant cells secreted more and differentially loaded Extracellular vesicles. These cargos include Wnt signaling proteins and miRNA targeting Wnt signaling components /target genes.

The data present is of high quality especially the characterization of EV and the miRNA and proteome profiles of these EVs. However, I feel that the functional validation of these findings and a biological significance is lacking in this manuscript and should be more of a focus for the authors.

A recent paper (Akella et al, Elife 2020 Feb 26;9:e50580. doi: 10.7554/eLife.50580.) describes the role of BBSome proteins in negatively regulateing EV shedding which seems to fit with the authors findings.

Figure 1: Why does neither GW4869 nor siAlix alter EV secretion in control cells? That seems to be odd? Is there another way to inhibit EV secretion from these cells?

Knockdown of Rab27 or HGS/Tsg101 or syntenin?

Minor comments:

The authors write about Wnt proteins being found on EV from ciliary mutant cells but for better understanding Wnt signaling proteins or components would be more appropriate.

The functional assays in Figure 5 are a good start, but I feel this is not thoroughly enough tested to claim the effects on Wnt signaling can be attributed to the Wnt signaling components and/or miRNA targeting Wnt signaling found on ciliary mutant EVs.

Figure 5:

Uptake assay:

Are EV from ciliary mutant cells taken up in a different manner, quantity?

WNT activity assay:

I find the Wnt activity effect presented not very convincing. In general, the dynamic range of the assay doesn't seem to be that big. In 4e the authors claim that due to the 4 fold increase of EV from the ciliary mutant cells, the effect would be much stronger. Can they perform Wnt activity assays with equal amounts of EV from all conditions. Is the proteins content fothose smEV pellets similar or different?

Have the authors considered using supernatant before and after 100.000g centrifugation to get rid of the EV-bound activity? This rescue of negative regulation of Wnt signaling in target cells would make it more convincing. In addition, another approach is necessary, such as quantitative PCR where Axin2 and other Wnt target genes should react to the treatment with EV from ciliary mutant cells.

What about the miRNA found? Some sort of rescue experiments that prove that the EV-bound Wnt components or miRNA targeting Wnt components are the causative agents of this activity.

Considering the Gskbeta-dependent integration of proteins into MVBs as mentioned by the authors. Maybe a functional effect on Wnt signaling is rather to be found in the EV producing cells instead of an intercellular communication? Can the authors test this hypothesis? For example, by using a Wnt reporter in the ciliary mutant and WT cells and compare the activity level with and without perturbation of MVB formation and EV secretion (GW4869, Alix, rab27)

Along these lines, what would be a biological explanation and context for Wnt packaging on EV in ciliated cells?

In general, this subject of this study is equally interesting for the EV and cilia community. For the field of Wnt signaling a better understanding of the purpose and effects of these changes in EV secretion in ciliary mutant cells would be necessary. Thus taken together, while I think this paper takes a good start on an interesting finding, I consider it yet too preliminary to justify publication in Nature Communications.

Reviewer #2 (Remarks to the Author):

NCOMMS-20-30089

Bardet-Biedl Syndrome-associated extracellular vesicles modulate WNT signalling

The manuscript by Votz et al studies the composition and function of extracellular vesicles shed into media of wild-type kidney medullary (KM) and Bardet-Biedl Syndrome mutant KM cell lines. Authors find that *Bbs4*^{-/-} and *Bbs6*^{-/-} cells produce four times as many EVs as wild-type KM cells, and the majority of small EVs; define the miRNA and protein cargoes of these EVs; discover that WNT pathway components are important EV cargoes that signal between cells. These data and these studies are high quality, rigorous, and grounded in the literature (ie do not over-interpret, of which many EVomics studies are prone). For example, EVs are characterized using multiple methods: biochemical, nanoparticle tracking analysis (NTA), and transmission electron microscopy (TEM). This work is impactful and important, given that cilia are found on most non-dividing cells in the human body and that cilia possess the conserved ability to shed extracellular vesicles, likely microvesicles/ectosomes from the ciliary tip, and here, authors propose, at the ciliary base/ciliary pocket at exosomes. The latter assertion is supported by the finding that exosomal markers are abundant in the small EV population that is shifted in that *Bbs4*^{-/-} and *Bbs6*^{-/-}. This work will appeal to the broad readership of Nature Communications. This reviewer has only the most minor of suggestions; no additional experiments are required.

Do *bbs4*^{-/-} and *bbs6*^{-/-} KM cells possess cilia? In lines 98-101, authors state "Loss of either gene results in ciliary trafficking defects and *Bbs4* and *Bbs6* KM cell lines have been extensively characterized and recapitulate the cilia phenotype of related cilia knockout or knockdown cell lines." For the non-expert, please explain the cilia phenotype and "related cilia KO or KD."

Line 66: Large EVs are described as 100-350nm. The terms small EVs (<200 nm) and large EV (>200 nm) are used in the EV field. References:

They, C. et al Minimal information for studies of extracellular vesicles 2018 (MISEV2018): a position statement of the International Society for Extracellular Vesicles and update of the MISEV2014 guidelines. *J Extracell Vesicles* 7, 1535750 (2018).

Supplementary Tables (with exception of ST1) are not labeled.

Authors are to be commended putting their studies in the context of all that is known about ciliary EVs and WNT signaling in EVs from multiple model systems. This speaks to the depth and breadth of knowledge of authors and to the major advance made by this work.

Reviewer #3 (Remarks to the Author):

Although the title of this paper is "Bardet-Biedl syndrome-associated extracellular vesicles modulate WNT signaling", there is no evidence for the role of BBS-associated EVs in modulation of WNT signaling in this paper. In addition, the argument is vague and unsettled in this paper.

Major points

1. The authors propose the title of this paper based mainly on the data shown in Fig. 5d, e. However, the data cannot support 'role of BBS-associated EVs in modulation of WNT signaling'. In Fig. 5d, there is no difference between WT and *Bbs4*^{-/-} or *Bbs6*^{-/-}. Furthermore, I cannot realize what the authors

intend to claim based on the data in Fig. 5e, nor understand what the last sentence in 'RESULT' section means as the sentence is ambiguous or may be misunderstanding. Compared with WT, whether do BBS-associated EVs regulate WNT signaling positively or negatively? 'Modulate' in the title is ambiguous.

2. In Fig. 4h, the two categories (positive and negative regulators of WNT signaling) are jumbled together as 'WNT proteins'. This concern is related to the above comment on Fig. 5d, e.

3. Many of the proteins listed in Fig. 4h as WNT proteins appear to be more general regulators, even though these are also involved in WNT signaling. For example, Psm proteins are subunits of the proteasome involved in general protein degradation; Snx3 and Vps35 are subunits of the retromer involved in retrograde trafficking of not only Wntless but also many other proteins; and Prkaa1 (AMP-kinase) and Gsk3beta are general kinases under the control of various pathways; in addition, Gsk3beta is found in EVs derived from cilia (see Ref. 23, Phua et al.). The authors should carefully discuss the possibility that these proteins participate in regulation of WNT signaling among various other regulatory pathways.

4. Fig. 1 lacks a critical control experiment that can discriminate the origins of EVs from cilia and non-cilia. The authors harvested EVs from ciliated WT, Bbs4^{-/-}, and Bbs6^{-/-} cells cultured under serum-starved conditions. However, the authors should compare the size (by NTA) and content (by LC-MS and small RNA-seq) between EVs prepared from ciliated (serum-starved) cells and non-ciliated (serum-fed) cells. Conditioned media can be harvested from cells cultured in medium containing exosome-depleted FBS (available from Gibco/Thermo Fisher etc.) and cells cultured under serum-starved conditions followed by addition of exosome-depleted FBS prior to EV preparation.

Manor points

1. Fig. 3a, b, and Fig. 4a, b, c: What does 'intensity' in the legend of the vertical axis mean?
2. Fig. 5e: What does 'impact' in the legend of the vertical axis mean? No error bars, no statistical analysis. There is no description about the used vesicle numbers or related information in the legend for Fig. 5d or e or in Materials and Methods; this minor point is related to the major point 1.
3. line 210: (Fig. 3 c,e)
4. line 222: Suppl. Table 3,4 and ?
5. line 288: therefore
6. line 398: using (NOT sing)

NCOMMS-20-30089

Bardet-Biedl Syndrome-associated extracellular vesicles modulate WNT signalling.

Reviewer #1 (Remarks to the Author):

The paper “Bardet-Biedl Syndrome-associated extracellular vesicles modulate WNT signaling” from Volz and colleagues describes the interesting finding that ciliary mutant cells secreted more and differentially loaded Extracellular vesicles. These cargos include Wnt signaling proteins and miRNA targeting Wnt signaling components /target genes.

The data present is of high quality especially the characterization of EV and the miRNA and proteome profiles of these EVs. However, I feel that the functional validation of these findings and a biological significance is lacking in this manuscript and should be more of a focus for the authors.

Response: We appreciate that the reviewer valued the quality of our work and have now expanded the results to include many more lines of experimentation described in more detail below. In brief we took a closer look at the origin of our additional vesicles, expanded the Wnt related findings and looked for similar trends in a clinically relevant patient derived cell line. We also want to highlight that one of the key findings of this paper is a shift in biogenesis and secretion of a distinct subset of molecules upon ciliary dysfunction. Lastly, we want to thank this reviewer for their helpful suggestions which we strongly believe have greatly improved the manuscript.

A recent paper (Akella et al, Elife 2020 Feb 26;9:e50580. doi: 10.7554/eLife.50580.) describes the role of BBSome proteins in negatively regulating EV shedding which seems to fit with the authors findings.

Response: We totally agree that this paper fits with our findings, although we had already referenced this manuscript in the previous version, we have now made a clearer statement highlighting these connections.

Figure 1: Why does neither GW4869 nor siAlix alter EV secretion in control cells? That seems to be odd? Is there another way to inhibit EV secretion from these cells? Knockdown of Rab27 or HGS/Tsg101 or syntenin?

Response: This is indeed somewhat surprising; it is possible that the inhibitor GW4869 and siAlix are only able to inhibit the additional EVs found in mutants. The inhibitor might be less active on more ‘stable’ cells, likewise, the effect of Alix knockdown might be more influential in the mutant cells. i.e. control cells might be better able to buffer homeostatic changes.

It must also be noted that the knockdown efficiency in these cells is not as high as with other cell types, as can be seen in Suppl. Fig 3. We attempted another method to reduce smEVs with KD of both Rab27a and Rab 35, but had extremely inefficient knock down.

To further validate the origin of our additional EVs we knocked down Annexin1, a protein known to be involved in IgEV biogenesis. After knock down of Annexin1 we saw no significant difference in the number of smEVs, supporting our hypothesis that extra EVs released from ciliary mutant cells are indeed MVB related.

To identify whether excess EVs were related to ciliary mechanism we repeated our original experiments with cells grown under serum-fed conditions. Addition of serum significantly reduced ciliation in ciliary mutant cells, and accordingly eliminated the additional EVs observed via NTA (see new Fig. 1 c-f). Addition of serum did not change the levels of ciliation in control cells, which remained around 55%. In serum-fed conditions all three cell lines released the same number of smEVs. We also performed LC-MS on these EV preparations. Upon serum-fed cells the smEV populations had far less proteins, most of which were also detected in serum-depleted smEV populations (see new Fig. 2 a). Although we cannot discriminate the origins of our smEVs it is clear that the excess smEVs found in ciliary mutant cells are generated as a consequence of ciliary dysfunction.

Minor comments:

The authors write about Wnt proteins being found on EV from ciliary mutant cells but for better understanding Wnt signaling proteins or components would be more appropriate.

Response: We tried to stay consistent and refer to proteins when dealing with the results from LC-MS, and miRNAs when dealing with the RNA sequencing data. We use components or molecules when referring to both.

The functional assays in Figure 5 are a good start, but I feel this is not thoroughly enough tested to claim the effects on Wnt signaling can be attributed to the Wnt signaling components and/or miRNA targeting Wnt signaling found on ciliary mutant EVs.

Response: We completely reworked Figure 5 and expanded it to a 6th figure. We performed further experiments to narrow down the origin of the Wnt response. We examined a clinically relevant patient derived cell line and found a similar trend and looked at the effect of secreting cells. More details are given in the responses below.

Figure 5:

Uptake assay: Are EV from ciliary mutant cells taken up in a different manner, quantity?

Response: This was one of the new experiments added to the manuscript ((Fig.5 a). Surprisingly, ciliary mutant EVs were less able to be taken up by our target cells. Particles were measured via NTA to ensure that the same number of particles were loaded. The fewer ciliary mutant smEVs were still able to illicit a response in target cells, which lead us to redesign our model in which the EVs secreted from ciliary mutant cells have a stronger effect (Fig.6 f)

WNT activity assay:

I find the Wnt activity effect presented not very convincing. In general, the dynamic range of the assay doesn't seem to be that big. In 4e the authors claim that due to the 4 fold increase of EV from the ciliary mutant cells, the effect would be much stronger. Can they perform Wnt activity assays with equal amounts of EV from all conditions. Is the proteins content of those smEV pellets similar or different?

Response: We have now completely reworked the data in figure 5. As the reviewer had suggested, we performed the luciferase assay with the same number of EVs. We observed a significant dampening

effect with all three EV samples. This data has to be considered alongside the uptake assay (described in the comment above). If we take into account that under physiological conditions ciliary mutant cells release many more EVs, the dampening effect inside target cells would be stronger. We have added this as a discussion point. We also determined that the effect was coming from smEVs, by isolated CD9/CD81/CD63-positive vesicles using an exosome isolation kit (via magnetic beads).

Have the authors considered using supernatant before and after 100.000g centrifugation to get rid of the EV-bound activity? This rescue of negative regulation of Wnt signaling in target cells would make it more convincing.

Response: Yes, we did indeed try to perform the luciferase assay with the supernatant before the 100.000g centrifugation step. Despite trying four times we were not able to get clean and consistent results. This was additionally hampered by the fact that NTA is not suitable to determine lgEV number, which makes normalization to EV number not possible.

However, to confirm that the WNT response was coming from smEVs, we isolated CD9/CD81/CD63-positive vesicles using an exosome isolation kit (via magnetic beads). CD9/CD81/CD63-positive vesicles were similarly able to dampen WNT activity in our reporter cell which implies that the dampening effect is largely a consequence of CD9/CD81/CD63-positive vesicles.

In addition, another approach is necessary, such as quantitative PCR where Axin2 and other Wnt target genes should react to the treatment with EV from ciliary mutant cells.

Response: We did attempt several different approaches but found that the Luciferase assay using the specialized TCF/LEF Reporter (luc)-HEK293 Cell Line was the most reliable and consistent. This was largely due to increased sensitivity of these cells to respond to WNT related changes. We got similar results using a Hek293T cell line transfected with reporter plasmids. But since this also depended on the efficiency of transfection, we switched to using a stably transfected reporter cell line. We found that WB & qPCR of target cells after EV application gave inconsistent results. We also tested different target cells, since we are conscious that HEK293T cells might not be the best target cell. Particularly since we are working across two different species. We do believe that our results demonstrate that our smEVs are able to illicit a response in target cells. Furthermore, as discussed in more detail below, we believe release of smEVs might also be affecting the WNT response in secreting cells.

What about the miRNA found? Some sort of rescue experiments that prove that the EV-bound Wnt components or miRNA targeting Wnt components are the causative agents of this activity.

Response: This was again a very interesting suggestion for which we now have additional data (Fig. 5 e). To determine whether the miRNA content found in our smEVs contributed to the Wnt activity in target cells we deplete smEVs of miRNAs, by knocking down Dicer in our KM cells prior to harvesting smEVs. Since Dicer knockdown prior to smEV harvesting made no difference on their ability to dampen the WNT response we propose that smEV protein content largely drives the responsiveness in this reporter cell line.

Considering the Gskbeta-dependent integration of proteins into MVBs as mentioned by the authors. Maybe a functional effect on Wnt signaling is rather to be found in the EV producing cells instead of an intercellular communication? Can the authors test this hypothesis? For example, by using a Wnt reporter in the ciliary mutant and WT cells and compare the activity level with and without perturbation of MVB formation and EV secretion (GW4869, Alix, rab27) Along these lines, what would be a biological explanation and context for Wnt packaging on EV in ciliated cells?

Response: We are glad that the reviewer pointed this out, since we do believe that the effect on Wnt might be on the secreting as well as the target cells. We checked for Wnt activity in the releasing cells using Cyclin-D1, a direct target of the Wnt signaling pathway and observed an upregulation of Wnt signaling in these ciliary mutant source cells. However, it is not clear whether the loading of more Wnt molecules in cilia mutant EVs might be causal or a consequence of disrupted Wnt signaling. The additional Wnt loaded EVs secreted by cilia mutant cells might be a negative secondary effect which could contribute to the disease phenotype. Of note, we also found other molecules (Hgs and Ykt6) known to play a role in sorting Wnt proteins for EV secretion, suggesting that this might be an active process. We have added this new data in Fig. 6, updated our model and also discussed this extensively in the discussion.

In general, this subject of this study is equally interesting for the EV and cilia community. For the field of Wnt signaling a better understanding of the purpose and effects of these changes in EV secretion in ciliary mutant cells would be necessary. Thus taken together, while I think this paper takes a good start on an interesting finding, I consider it yet too preliminary to justify publication in Nature Communications.

Response: We are glad that the reviewer could see the potential in our work and that it could be of benefit to such a wide-ranging scientific community. We are indeed very grateful for all the helpful suggestions and hope that they appreciate all the additional experiments that we believe massively strengthen our initial findings. We were particularly excited to see that our initial findings were recapitulated in a BBS patient cell line.

Reviewer #2 (Remarks to the Author):

The manuscript by Votz et al studies the composition and function of extracellular vesicles shed into media of wild-type kidney medullary (KM) and Bardet-Biedl Syndrome mutant KM cell lines. Authors find that Bbs4^{-/-} and Bbs6^{-/-} cells produce four times as many EVs as wild-type KM cells, and the majority of small EVs; define the miRNA and protein cargoes of these EVs; discover that WNT pathway components are important EV cargoes that signal between cells. These data and these studies are high quality, rigorous, and grounded in the literature (ie do not over-interpret, of which many EVomics studies are prone). For example, EVs are characterized using multiple methods: biochemical, nanoparticle tracking analysis (NTA), and transmission electron microscopy (TEM). This work is impactful and important, given that cilia are found on most non-dividing cells in the human body and that cilia possess the conserved ability to shed extracellular vesicles, likely microvesicles/ectosomes from the ciliary tip, and here, authors propose, at the ciliary base/ciliary pocket at exosomes. The latter assertion is supported by the finding that exosomal markers are abundant in the small EV population that is shifted in that Bbs4^{-/-} and Bbs6^{-/-}. This work will appeal to the broad readership of Nature Communications. This reviewer has only the most minor of suggestions; no additional experiments are required.

Response: We are incredibly grateful that this reviewer was able to appreciate the value of our work. We hope that they are equally satisfied with the additional experiments that we performed to address the other Reviewers comments. We are particularly pleased that we were able to add some data from a BBS patient cell line that recapitulated our findings.

Do bbs4^{-/-} and bbs6^{-/-} KM cells possess cilia? In lines 98-101, authors state “Loss of either gene

results in ciliary trafficking defects and Bbs4 and Bbs6 KM cell lines have been extensively characterized and recapitulate the cilia phenotype of related cilia knockout or knockdown cell lines.” For the non-expert, please explain the cilia phenotype and “related cilia KO or KD.”

Response: We have now included this data to show that upon serum starvation Bbs4^{-/-} and Bbs6^{-/-} KM cells displayed significantly more ciliated cells with significantly longer cilia. This has been added as a new supplementary figure (Supp. Fig.1). We also looked at the state of ciliation when these cells were grown with the presence of serum, which reduced the level of ciliation in mutants, but not in control cells.

Query: Line 66: Large EVs are described as 100-350nm. The terms small EVs (<200 nm) and large EV (>200 nm) are used in the EV field. References: They, C. et al Minimal information for studies of extracellular vesicles 2018 (MISEV2018): a position statement of the International Society for Extracellular Vesicles and update of the MISEV2014 guidelines. J Extracell Vesicles 7, 1535750 (2018).

Response: We have updated this and included the reference. Thank you for pointing this out.

Supplementary Tables (with exception of ST1) are not labeled.

Response: We apologize for how this was displayed in the submission. We had originally included these tables as Excel Files and didn't expect them to be listed as shown. We have updated the labels and will now include this as supplementary data.

Authors are to be commended putting their studies in the context of all that is known about ciliary EVs and WNT signaling in EVs from multiple model systems. This speaks to the depth and breadth of knowledge of authors and to the major advance made by this work.

Response: Thank you for these words of encouragement.

Reviewer #3 (Remarks to the Author):

Although the title of this paper is “Bardet-Biedl syndrome-associated extracellular vesicles modulate WNT signaling”, there is no evidence for the role of BBS-associated EVs in modulation of WNT signaling in this paper. In addition, the argument is vague and unsettled in this paper.

Response: We thank the reviewer for their time and constructive criticism which we have taken to heart and incorporated into this new submission. We have now expanded the results to include many more lines of experimentation described in more detail below. In brief we took a closer look at the origin of our additional vesicles, expanded the Wnt related findings and looked for similar trends in a clinically relevant patient derived cell line. We also want to highlight that one of the key findings of this paper is a shift in biogenesis and secretion of a distinct subset of molecules upon ciliary dysfunction. Therefore, we have changed the title to better reflect the main message of the paper and hope to highlight that loss of BBS proteins result in a change of biogenesis and release of bioactive molecules. The new title is now ‘Bardet-Biedl Syndrome Proteins Modulate the Release of Bioactive Extracellular Vesicles.

Major points

1. The authors propose the title of this paper based mainly on the data shown in Fig. 5d, e. However, the data cannot support 'role of BBS-associated EVs in modulation of WNT signaling'. In Fig. 5d, there is no difference between WT and Bbs4^{-/-} or Bbs6^{-/-}. Furthermore, I cannot realize what the authors intend to claim based on the data in Fig. 5e, nor understand what the last sentence in 'RESULT' section means as the sentence is ambiguous or may be misunderstanding. Compared with WT, whether do BBS-associated EVs regulate WNT signaling positively or negatively? 'Modulate' in the title is ambiguous.

Response: We agree that in the initial submission the data in figure 5 needed strengthening. We have now completely reworked the data in figure 5 and expanded this to a 6th figure.

We repeated the uptake assay to include ciliary mutant EVs and showed that these were less able to be taken up by our target cells (Fig.5 a). Particles were measured via NTA to ensure that the same number of particles were loaded. We also repeated the luciferase assay using the same number of EVs upon which we observed a significant dampening effect with all three EV samples (Fig.5 c). This data has to be considered alongside the uptake assay. Since fewer ciliary mutant smEVs were still able to illicit a response in target cells, which lead us to redesign our model in which the EVs secreted from ciliary mutant cells have a stronger effect (Fig.6 f). If we take into account that under physiological conditions ciliary mutant cells release many more EVs, the dampening effect inside target cells would be stronger. This has been added to the discussion.

We also determined that the effect was coming from smEVs, by isolated CD9/CD81/CD63-positive vesicles using an exosome isolation kit (via magnetic beads) and less likely from the miRNA (Fig.5 d, e). We were particularly excited to see that our initial findings were recapitulated in a BBS patient cell line.

We also changed the title of the manuscript as described above.

2. In Fig. 4h, the two categories (positive and negative regulators of WNT signaling) are jumbled together as 'WNT proteins'. This concern is related to the above comment on Fig. 5d, e.

Response: We agree that this is intriguing. However, these are the results that we obtained from the GetGo analysis that we displayed in an unbiased manner. We believe that it might reflect the ability of smEV release to influence not only the target cells, but also the secreting cells. We have included this explanation in the discussion.

3. Many of the proteins listed in Fig. 4h as WNT proteins appear to be more general regulators, even though these are also involved in WNT signaling. For example, Psm proteins are subunits of the proteasome involved in general protein degradation; Snx3 and Vps35 are subunits of the retromer involved in retrograde trafficking of not only Wntless but also many other proteins; and Prkaa1 (AMP-kinase) and Gsk3beta are general kinases under the control of various pathways; in addition, Gsk3beta is found in EVs derived from cilia (see Ref. 23, Phua et al.). The authors should carefully discuss the possibility that these proteins participate in regulation of WNT signaling among various other regulatory pathways.

Response: We defined the Wnt proteins as described in UniProt. We agree with this point and have included it in the discussion. Our discussion is now expanded and also highlights the other signaling pathways for which we found differentially enriched molecules.

4. Fig. 1 lacks a critical control experiment that can discriminate the origins of EVs from cilia and non-cilia. The authors harvested EVs from ciliated WT, Bbs4^{-/-}, and Bbs6^{-/-} cells cultured under serum-starved conditions. However, the authors should compare the size (by NTA) and content (by LC-MS and small RNA-seq) between EVs prepared from ciliated (serum-starved) cells and non-ciliated (serum-fed) cells. Conditioned media can be harvested from cells cultured in medium containing exosome-depleted FBS (available from Gibco/Thermo Fisher etc.) and cells cultured under serum-starved conditions followed by addition of exosome-depleted FBS prior to EV preparation.

Response: This was an excellent suggestion. We acquired this medium and repeated the experiments with cells grown under serum-fed conditions. Addition of serum significantly reduced ciliation in ciliary mutant cells, and accordingly eliminated the additional EVs observed via NTA (see new Fig. 1 c-f). Addition of serum did not change the levels of ciliation in control cells, which remained around 55%. In serum-fed conditions all three cell lines released the same number of sEVs. We also performed LC-MS on these EV preparations. Upon serum-fed cells the smEV populations had far less proteins, most of which were also detected in serum-depleted smEV populations (see new Fig. 2 a). Although we cannot discriminate the origins of our smEVs it is clear that the excess smEVs found in ciliary mutant cells are generated as a consequence of ciliary dysfunction.

Manor points

1. Fig. 3a, b, and Fig. 4a, b, c: What does ‘intensity’ in the legend of the vertical axis mean?

Response: The intensity is the sum of all raw intensity measurements for each protein. It is only used for illustration of the scatter plots. Ratios are calculated from LFQ intensities. We have updated this in the methods section.

2. Fig. 5e: What does ‘impact’ in the legend of the vertical axis mean? No error bars, no statistical analysis. There is no description about the used vesicle numbers or related information in the legend for Fig. 5d or e or in Materials and Methods; this minor point is related to the major point 1.

Response: As we discussed above, we completely reworked the original figure 5. We repeated the original experiments and added several new ones. We agree that our original figure 5e was not satisfactory, and instead we repeated the experiment with loading the same number of EVs derived from mutant and control cells. This data must also be interpreted alongside the new finding that mutant EVs are taken up less than control EVs, yet still eliciting a similar response.

3. line 210: (Fig. 3 c,e)

Response: This has been amended.

4. line 222: Suppl. Table 3,4 and ?

Response: This has been amended.

5. line 288: therefore

Response: With the addition of new data, this section has been completely restructured.

6. line 398: using (NOT sing)

Response: We have completely rewritten the discussion and this sentence has now been removed.

REVIEWERS' COMMENTS

Reviewer #1 (Remarks to the Author):

In my view, Volz and colleagues have greatly improved their manuscript „Bardet-Biedl Syndrome Proteins Modulate the Release of Bioactive Extracellular Vesicles“ by additional experiments to show cell autonomous effects of active EVs, text editing and more precise interpretation.

I still see some weakness in the Wnt signaling effect of these EVs. Why did the Topflash assay or Axin 2 level not work in the Bbs4/6 KO cells? Could the authors show the autocrine effect of Wnt signaling components on LRP6 phosphorylation and beta-catenin stabilisation, which is a straight-forward assay or is the level of proteasomes decreased in those cells? This would also be helpful to show in the patient BBS10 and control cells.

Minor points:

There are still several text passages where the authors refer to Wnt proteins, this should be corrected.

Line 219 individual Wnt proteins – means the ligand, rather proteins of the Wnt signaling pathway, Wnt related proteins etc.

Line 250, 251, 257 not Wnt proteins - proteins of the Wnt signaling pathway or Wnt signaling proteins

Line 310 error in 30%

Line 335 should refer to Fig5h

Reviewer #3 (Remarks to the Author):

This paper has been improved almost to my satisfaction.

Draft Only

Reviewer #1 (Remarks to the Author):

In my view, Volz and colleagues have greatly improved their manuscript „Bardet-Biedl Syndrome Proteins Modulate the Release of Bioactive Extracellular Vesicles“ by additional experiments to show cell autonomous effects of active EVs, text editing and more precise interpretation.

I still see some weakness in the Wnt signaling effect of these EVs. Why did the Topflash assay or Axin 2 level not work in the Bbs4/6 KO cells? Could the authors show the autocrine effect of Wnt signaling components on LRP6 phosphorylation and beta-catenin stabilisation, which is a straight-forward assay or is the level of proteasomes decreased in those cells? This would also be helpful to show in the patient BBS10 and control cells.

Response: We are pleased that the reviewer could appreciate the additional experimentation and updated manuscript. The TopFlash assays performed in this manuscript were done using a stably transfected TCF/LEF HEK293T reporter cell line. Previous experiments using transfected HEK293T cell lines were found to be extremely variable and inconsistent since transfection of multiple reporter constructs are required. Given that the Bbs4/6 KO cells exhibit low transfection efficiency, performing this assay would yield unreliable results.

Numerous papers have already shown that suppression of BBS proteins leads to a stabilization of beta-catenin and perturbed proteasomal degradation. References for these are given in the manuscript. As I am sure the reviewer is aware, beta-catenin is a complex molecule and activation, or inactivation is dependent on various post translational modifications. Despite testing many different beta-catenin antibodies we found high levels of inconsistency between Western Blot and Immunocytochemistry. Therefore, we chose to use expression of Cyclin D1 as a reliable downstream target. These cells have already been previously characterized and have shown to exhibit WNT related defects (see Hernandez-Hernandez et al., HMG 2013). Furthermore, detailed characterization of WNT signaling components in these and other patient UREC cell lines are currently ongoing, yet beyond the scope of this manuscript.

We have updated the discussion to include this point.

Minor points:

There are still several text passages where the authors refer to Wnt proteins, this should be corrected.

Line 219 individual Wnt proteins – means the ligand, rather proteins of the Wnt signaling pathway, Wnt related proteins etc.

Line 250, 251, 257 not Wnt proteins - proteins of the Wnt signaling pathway or Wnt signaling proteins.

Response: Thank you for this observation we have gone through the text and changed all instances of incorrect wording.

Line 310 error in 30%

Response: This is a tilde character and became unformatted during generation of the PDF document.

Line 335 should refer to Fig5h

Response: This has been amended.

Reviewer #3 (Remarks to the Author):

This paper has been improved almost to my satisfaction.

Response: We are pleased that we could address this reviewers' comments and thank them for their helpful suggestions, which we believe greatly improved the manuscript.